# COLOR3D: CONTROLLABLE AND CONSISTENT 3D COLORIZATION WITH PERSONALIZED COLORIZER

**Yecong Wan**[12]**, Mingwen Shao**[3*]**, Renlong Wu**[1]**, Wangmeng Zuo**[1*]
[1]Faculty of Computing, Harbin Institute of Technology
[2]Harbin Institute of Technology Zhengzhou Research Institute
[3]Artificial Intelligence Research Institute, Shenzhen University of Advanced Technology
{yecongwan,hirenlongwu}@gmail.com,mw278@126.com,wmzuo@hit.edu.cn

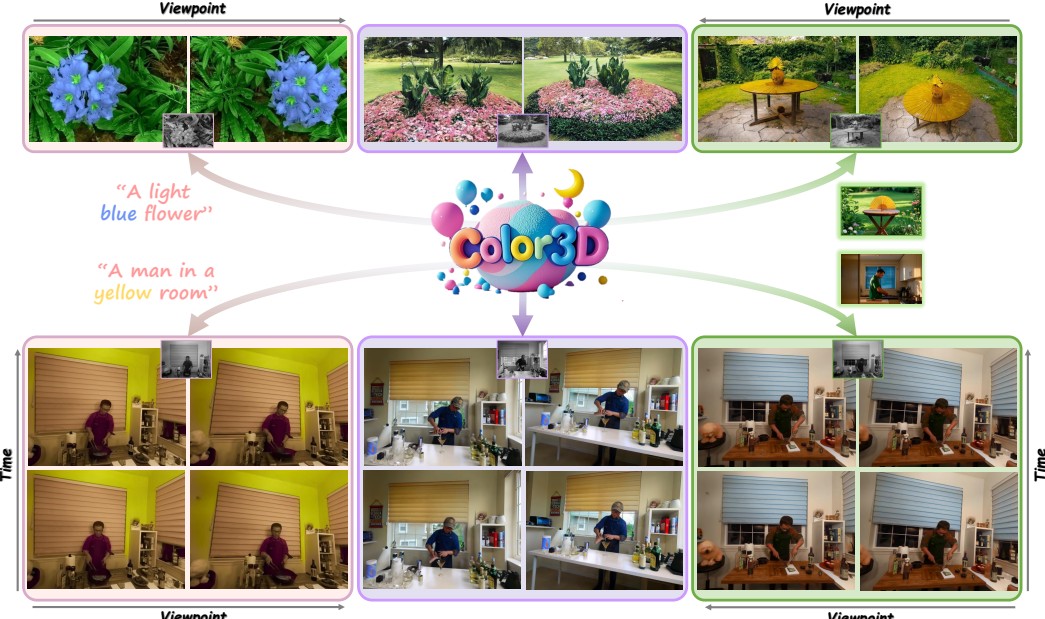

Figure 1: **Exemplary Visual Results of Color3D**. Color3D is a unified controllable 3D colorization framework for both static and dynamic scenes, producing vivid and chromatically rich renderings with strong cross-view and cross-time consistency. Our method supports diverse colorization controls, including language-guided (left), automatic inference (middle), and reference-based (right), showcasing its versatility and practical value.

## ABSTRACT

In this work, we present **Color3D**, a highly adaptable framework for colorizing both static and dynamic 3D scenes from monochromatic inputs, delivering visually diverse and chromatically vibrant reconstructions with flexible user-guided control. In contrast to existing methods that focus solely on static scenarios and enforce multi-view consistency by averaging color variations which inevitably sacrifice both chromatic richness and controllability, our approach is able to preserve color diversity and steerability while ensuring cross-view and cross-time consistency. In particular, the core insight of our method is to colorize only a single key view and then fine-tune a personalized colorizer to propagate its color to novel views and time steps. Through personalization, the colorizer learns a scene-specific deterministic color mapping underlying the reference view, enabling it to consistently project corresponding colors to the content in novel views and video frames via its inherent inductive bias. Once trained, the personalized colorizer can be applied to infer consistent chrominance for all other images, enabling direct reconstruction of colorful 3D scenes with a dedicated Lab color space Gaussian splatting representation. The proposed framework ingeniously recasts

---

*Corresponding Author

complicated 3D colorization as a more tractable single image paradigm, allowing seamless integration of arbitrary image colorization models with enhanced flexibility and controllability. Extensive experiments across diverse static and dynamic 3D colorization benchmarks substantiate that our method can deliver more consistent and chromatically rich renderings with precise user control. Project Page: `https://yecongwan.github.io/Color3D/`.

# 1 INTRODUCTION

The emerging 3D reconstruction techniques, such as Neural Radiance Fields (NeRF) Mildenhall et al. (2021) and 3D Gaussian Splatting (3DGS) Kerbl et al. (2023), have catalyzed high-fidelity novel-view synthesis from the observed color images. These methods evolves rapidly Chen et al. (2022); Fridovich-Keil et al. (2022); Garbin et al. (2021); Müller et al. (2022); Yu et al. (2024); Cheng et al. (2024a); Wu et al. (2024); Lu et al. (2024), and are extended to dynamic scenes Liu et al. (2023b); Guo et al. (2023); Shao et al. (2023); Duan et al. (2024); Li et al. (2024b); Yang et al. (2023); Wu et al. (2024); Yang et al. (2024c) with object motion modeling. Despite these advances, there is still a compelling challenge, i.e., reconstructing colorful 3D scenes from monochromatic inputs. This can significantly enhance the visual realism and expression of 3D reconstructions, unlocking new applications across digital art, artistic creation, and cultural heritage preservation.

In light of this, a straightforward way is to train a 3D colorization model on massive collections of 3D scene data. However, the formidable complexity of 3D geometry Lu et al. (2024), and the temporal intricacies in dynamic scenes Yan et al. (2024), making it prohibitively infeasible in practice. An alternative solution is to first colorize multi-view monochromatic images with 2D colorization models and then reconstruct the 3D content. Nevertheless, its naive implementation leads to severe cross-view color shift, due to latest 2D colorization models Kang et al. (2023); Yang et al. (2024a) struggling to colorize multi-view images consistently (See Fig. 18).

To mitigate the color inconsistency, a few attempts Dhiman et al. (2023); Cheng et al. (2024b) aim to average multi-view color variations via distillation Dhiman et al. (2023) or dynamic color injection Cheng et al. (2024b). While partially effective, they inevitably dilute the palette richness, producing desaturated and tone-flattened results. Furthermore, smoothing color variations makes the colorized results unpredictable, sacrificing user-controlled colorization ability. Moreover, existing studies only focus on static scenes, while controllable colorization of dynamic ones remains an open and unexplored problem, where color consistency in spatial and temporal dimensions should be maintained.

In this work, we suggest a novel paradigm for 3D colorization, i.e., first colorizing a key image and then propagating the color information to novel views and time steps by fine-tuning a personalized colorizer. The benefits are three-fold. First, it transforms the complex 3D colorization into a simpler color information propagation task, enabling more consistent results. Second, since only a key image needs to be controlled, it is easier to achieve controllable 3D colorization. Third, it provides a chance for unifying 3D colorization of both static and dynamic scenes.

Driven by the motivations, we propose **Color3D**, a unified controllable 3D colorization framework for both static and dynamic scenes (Fig. 1), where a personalized colorizer is optimized for each scene to enable consistent color propagation. Specifically, Color3D first selects the most informative view from the monochromatic images, and then colorizes it with a desirable off-the-shelf image colorization model. Next, to colorize other views or timesteps consistently, Color3D personalizes a scene-specific colorizer to learn the deterministic color mapping underlying this colorized view. In which a single view augmentation strategy is devised to expand the sample space, thereby enhancing the colorizer's generalization and its capacity to handle unseen visual content. Finally, the personalized colorizer is employed to infer consistent chromatic content of remaining views or frames, followed by direct reconstruction of colorful static or dynamic 3D scenes via a dedicated Lab color space Gaussian splatting representation, where luminance and chrominance components are optimized separately. Extensive experiments on public static and dynamic 3D datasets substantiate that our Color3D can produce colorful 3D content that aligns with user intentions, outperforming alternatives quantitatively and qualitatively. Additionally, Color3D also achieves visually promising results on real-world in-the-wild scenes, further demonstrating the practicality of our method in legacy restoration.

In conclusion, the main contributions are summarized as follows:

- We propose a unified and versatile 3D colorization framework, termed Color3D, which enables user-guided colorization of both static and dynamic scenes by tuning a personalized colorizer for each scene, advancing the controllability and interactivity of 3D colorization.
- To facilitate personalized colorizer tuning, we propose a customized key view selection strategy along with a single view augmentation scheme to enhance coloring richness and generalization. Furthermore, we introduce a Lab Gaussian representation to improve color reconstruction fidelity and better preserve scene structures.
- Our Color3D delivers vibrant, controllable, consistent results that faithfully align with user intent, significantly outperforming alternatives on a variety of benchmarks.

## 2 RELATED WORK

**Radiance Fields for Static and Dynamic Scenes.** Radiance field representations Mildenhall et al. (2021); Kerbl et al. (2023) have driven a paradigm shift in novel view synthesis. Neural Radiance Fields (NeRF) Mildenhall et al. (2021) pioneered implicit volumetric modeling with coordinate-based neural networks, inspiring numerous extensions Barron et al. (2021; 2022; 2023); Verbin et al. (2022); Chen et al. (2022); Fridovich-Keil et al. (2022); Garbin et al. (2021); Müller et al. (2022) and subsequent work on dynamic scenes Park et al. (2021a;b); Wang et al. (2023); Fang et al. (2022); Liu et al. (2023b); Guo et al. (2023); Shao et al. (2023). However, NeRF-based methods remain hindered by costly training and slow rendering. To overcome this, 3D Gaussian Splatting (3DGS) Kerbl et al. (2023) introduced an explicit Gaussian representation that enables real-time, photorealistic rendering. Recent advances further extend Gaussian splatting to dynamic reconstruction Duan et al. (2024); Li et al. (2024b); Yang et al. (2023); Wu et al. (2024); Yang et al. (2024c), combining efficiency and fidelity. Motivated by these developments, we employ 3DGS and 4DGS as canonical static and dynamic representations, building a unified framework for controllable 3D colorization with strong spatial-temporal consistency.

**From 2D Colorization to 3D.** Image colorization aims to recover plausible chromatic content from grayscale inputs, with decades of progress improving realism and controllability Zhang et al. (2017); Wu et al. (2021); Kim et al. (2022); Kang et al. (2023); Ji et al. (2022). Beyond automatic approaches, user-guided methods leverage reference images He et al. (2018); Huang et al. (2022); Zhao et al. (2021) or language prompts Weng et al. (2023); Zabari et al. (2023); Weng et al. (2022); Chang et al. (2023); Liang et al. (2024). Extending to videos introduces temporal consistency challenges, tackled by matching Yang et al. (2024b), palette transfer Wang et al. (2025), attention Li et al. (2024a), and memory propagation Yang et al. (2024a). Yet these remain limited to 2D domains, lacking cross-view consistency. Efforts on 3D scene colorization are still nascent. GBC Liao et al. (2024) exploits video models for continuous inputs, while ChromaDistill Dhiman et al. (2023) and ColorNeRF Cheng et al. (2024b) transfer knowledge from pretrained colorizers by averaging inconsistent predictions, often sacrificing controllability and vividness. Moreover, colorizing dynamic 3D scenes while ensuring spatial-temporal consistency remains unaddressed. Our work bridges this gap with a unified framework for both static and dynamic settings, achieving visually rich, consistent, and user-controllable 3D colorization.

## 3 METHODOLOGY

Our core idea is to fine-tune a personalized colorizer for each scene using a single colorized key view and then employ it to propagate the color information to novel views and time steps. By specializing in the scene-specific deterministic color mapping derived from this reference view, the colorizer is able to project the same content in novel views to the corresponding colors in the reference view through the inherent inductive bias capability, thereby elegantly addressing the challenge of color consistency across both static and dynamic 3D scenes.

### 3.1 OVERVIEW OF COLOR3D

The schematic illustration of the proposed Color3D is depicted in Fig. 2. Color3D is primarily composed of two phases, i.e., the personalized colorizer training stage and the 3D scene colorization stage. In the first stage, we begin by selecting a single key view from the inputs and apply an off-the-shelf image colorization model to generate a colorized reference view. With the augmented data from a customized single view augmentation scheme, we then fine-tune a personalized colorizer to learn the definite color mapping underlying this reference view. In the second stage, the personalized colorizer is employed to infer consistent chromatic content for the remaining views or frames, which are directly leveraged to optimize the vibrant 3D scene with a tailored Lab Gaussian representation.

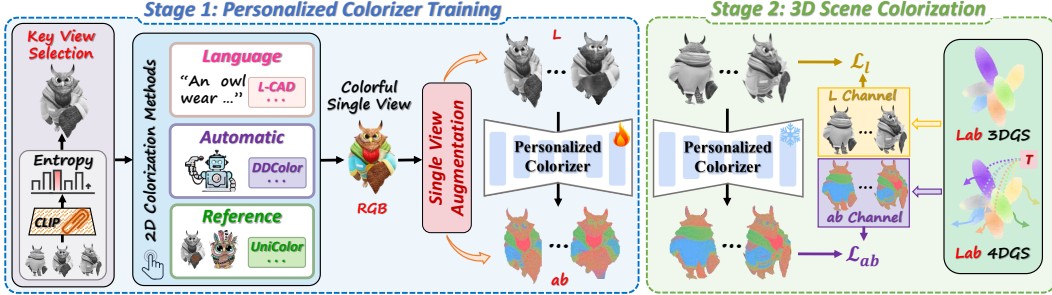

Figure 2: The overall pipeline of Color3D. Our framework comprises two primary stages. In the first stage, we initially identify the most informative key view from the given monochromatic images and video frames, and employ an off-the-shelf image colorization model to generate a colorized single view. Then, a single view augmentation scheme is elaborated to amplify the data, and the augmented samples are subsequently used to fine-tune a per-scene personalized colorizer. In the second stage, this personalized colorizer is utilized to infer consistent chromatic content of the remaining views or frames, and directly reconstruct the colorful 3D scene with Lab color space 3DGS or 4DGS.

## 3.2 STAGE 1: PERSONALIZED COLORIZER TRAINING

In the initial stage, we aim to fine-tune a personalized colorizer using a single colorized reference view, which enables consistent color propagation to novel views and frames. In contrast to training a standard image colorization model on diverse samples with inherent one-to-many color mappings, which results in inconsistent color predictions under minor contextual changes, our personalized colorizer is explicitly trained to learn the scene-specific one-to-one color mapping derived from a single image. This allows for consistent chromatic outputs across both spatial and temporal variations.

**Key View Selection.** Since our goal is to fine-tune a personalized colorizer using a single view, this view should capture the widest range of visual information while minimizing redundancy, which is crucial to ensure that the colorizer can be more robust to handle other views of the scene. To achieve this, we propose a heuristic strategy that selects the most informative and representative view by comparing feature similarity entropies across all candidates.

Specifically, given a set of $N$ candidate monochromatic inputs, we first extract their single dimension feature representations using CLIP Radford et al. (2021) and the concated feature matrix can be denoted as $F \in \mathbb{R}^{N \times D}$, where $D$ is the dimension of the feature space. To ensure that similarity comparisons reflect angular distances, we normalize each feature vector to unit length $\hat{F}_i = \frac{F_i}{\|F_i\|_2}$, for $i = 1, 2, \ldots, N$. Then, we compute the pairwise cosine similarity matrix to measure the relationships between different views $S = \hat{F}\hat{F}^T$, where $S_{ij}$ represents the similarity between view $i$ and view $j$. For each view $i$, we define a probability distribution $P_{ij}$ over its similarity with all other views, and use this distribution to compute the information entropy $H(I_i)$ associated with view $i$:

$$H(I_i) = -\sum_j P_{ij} \log P_{ij}, \qquad P_{ij} = \frac{\exp(S_{ij})}{\sum_k \exp(S_{ik})} \qquad (1)$$

A higher entropy value indicates that the view is more evenly related to all other views, meaning it encapsulates a broader and more diverse set of visual information. We define the optimal key view as the one that maximizes the entropy:

$$I^* = \arg\max_{I_i} H(I_i). \qquad (2)$$

**Key View Colorization.** After obtaining the key view, users can apply any off-the-shelf image colorization model to perform colorization on the single view. Such a single-view-based strategy allows for the seamless integration of various mature techniques, including language-guided Weng et al. (2023); Chang et al. (2023); Liang et al. (2024), reference-based He et al. (2018); Huang et al. (2022), and automatic Kang et al. (2023); Ji et al. (2022) image colorization methods without any additional tuning or modification, thereby offering more flexible and convince controllability over both static and dynamic 3D scene colorization.

**Single View Augmentation.** In order to avoid overfitting and more robustly generalize to novel views and video frames, we propose a single view augmentation scheme that combines generative augmentations and traditional augmentations to generate more samples in which the same content consistently retains its color across all instances. As illustrated in Fig. 3(a), the proposed scheme

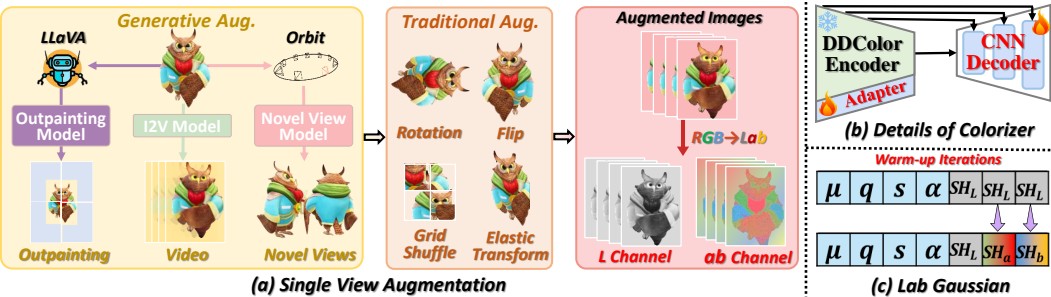

Figure 3: (a): Illustration of the proposed single view augmentation scheme that combines generative augmentations and traditional augmentations to enrich the single colored view with consistent color distribution. (b): Architecture of the colorizer consists of a frozen DDColor encoder alongside a trainable adapter and CNN decoder. (c): Lab Gaussian that first warms up with three $L$ channels and then switches to full $Lab$ channels for color optimization.

first leverages generative models to produce plausible scene content beyond the provided single view, while ensuring chromatic coherence with the input to enrich the diversity of colorization supervision. Specifically, given a single colored view, we incorporate three complementary generative strategies: (1) Outpainting: we divide the image into a $2 \times 2$ grid and apply Stable Diffusion Rombach et al. (2022) to each of the four regions individually to imagine plausible scene content beyond their outer boundaries. Meanwhile, we employ LLaVA Liu et al. (2023a) to generate a descriptive caption based on the image, which serves as the text prompt to guide the diffusion process. (2) Image-to-Video: to simulate newly appearing objects and motions in dynamic scenes, we generate continuous video frames conditioned on a single image using Stable Video Diffusion Blattmann et al. (2023). This approach provides coherent object motions and gradual changes in viewpoint, enabling the personalized colorizer to learn more robust color assignments for content that emerges over time. (3) Novel View: to simulate consistent novel viewpoints, we employ Stable Virtual Camera Zhou et al. (2025) to generate views along a predefined orbital trajectory with consistent content across perspectives. Notably, we do not require the generated content to be completely the same as the scene to be colored; instead, we emphasize preserving consistent chromatic styles with the input view, which is sufficient to improve coloring robustness and enhance coloring richness for scene content not included in the single view.

Afterward, we apply a series of traditional augmentations to further enhance the samples, including rotation, flip, grid shuffle that divides the image into grids and randomly disrupts them, and elastic transform Simard et al. (2003) that introduces smooth deformations to simulate realistic variations in object shape and structure. This can facilitate the learning of more robust and variation-agnostic color mapping.

Finally, these augmented images are converted from RGB to the CIE Lab color space Iizuka et al. (2016), where the $L$ channel is used as the input to the colorizer and the $ab$ channels serve as the target for color prediction. The Lab color space not only decouples luminance from chromatic components but also offers a perceptually uniform representation that better aligns with human visual perception.

**Colorizer Tuning.** With the augmented samples, we can fine-tune a personalized colorizer to learn the scene-specific, variation-agnostic, deterministic color mapping underlying the single view. The structure of the colorizer is illustrated in Fig. 3(b). We adopt an encoder from the pre-trained image colorization model DDColor Kang et al. (2023) and freeze its parameters to preserve its strong capability in extracting high-level color-relevant features. This enables our model to learn semantic-level color mappings from limited samples, which are inherently more robust to viewpoint and motion variations than low-level luminance-to-color correspondences. On the other hand, we deliberately avoid using a pre-trained decoder, whose built-in color priors are a major source of multi-view color inconsistency. Instead, we employ a lightweight, learnable CNN-based decoder initialized from scratch to serve as a clean and unbiased slate for learning personalized color mappings. Furthermore, we equip the encoder with additional learnable adapters Chen et al. (2024a) to better adapt to the current scenario while preserving its original pre-trained priors. The personalized colorizer is fine-tuned using a simple L1 loss:

$$\mathcal{L}_{colorizer} = \quad \| P^{ab} - G^{ab} \|_1 \tag{3}$$

where $P^{ab}$ and $G^{ab}$ denote the predicted $ab$ channels by the personalized colorizer and the ground-truth $ab$ channels, respectively. Please refer to Appendix A.2 for additional network details and a more in-depth explanation of why the personalized colorizer can achieve consistent color propagation.

### 3.3 STAGE 2: 3D SCENE COLORIZATION

With the personalized colorizer, in this stage, we aim to generate consistent colors for other views and video frames, which are then directly utilized to reconstruct colorful static and dynamic 3D scenes. Considering that the luminance information of the scene is already known and it can serve as reliably scene structure constraint, we advocate optimizing in CIE Lab color space Iizuka et al. (2016). By decoupling the known luminance ($L$ channel) from the predicted chrominance ($ab$ channels), our approach can constrain them separately to more stably optimize the 3D scene, attenuating the effect of perturbations in the predicted chromatic information. To achieve this, we deploy 3DGSKerbl et al. (2023) and 4DGS Wu et al. (2024) to model static and dynamic 3D scenes and propose a Lab Gaussian representation for rendering images in Lab color space and facilitating separate optimization of luminance and chrominance.

**Lab Gaussian.** The vanilla Gaussian attributes including position ($\mu$), rotation ($q$), scaling ($s$), opacity ($\alpha$), and Spherical Harmonics (SH) coefficients ($SH_R, SH_G, SH_B$) for modeling RGB color channels. To better deouple luminance and chrominance, we retain the same architecture but reformulate the three sets of SH coefficients to the Lab color space as $\{SH_L, SH_a, SH_b\}$, which separately parameterize the luminance ($L$) and chrominance ($a$ and $b$) channels. Following the original splatting pipeline Kerbl et al. (2023), we can render images with Lab color space representation directly from these SH coefficients. This strategy allows for fully leveraging the high-fidelity luminance information to provide more stable optimization signals for Gaussian primitives such as position and shape, thereby improving training stability and scene convergence.

Due to the differing numerical ranges between $L$ and $ab$ channels in the Lab color space, directly modeling chrominance and luminance jointly with the same Gaussian primitives can lead to unstable optimization and convergence. To mitigate this issue, we normalize rendered Lab channels into a unified range of $[0, 1]$. Specifically, the luminance channel $L$, originally in $[0, 100]$, is scaled by a factor of $1/100$. The chrominance channels $a$ and $b$, originally spanning $[-128, 127]$, are first shifted by 128 and then scaled by $1/255$ accordingly. The normalization process can be formulated as:

$$L' = \frac{L}{100}, \quad a' = \frac{a + 128}{255}, \quad b' = \frac{b + 128}{255} \tag{4}$$

**Optimization objectives.** We deploy the same loss functions for both static and dynamic scenarios and otherwise remain identical with the original 3DGS and 4DGS. For the rendered image in the Lab color space, we optimize the $L$ and $ab$ channels separately. Since the $L$ channel contains the core structural and high-frequency detail information of the scene, we deploy an edge loss $\mathcal{L}_{\text{edge}}$ to encourage the Gaussian primitives to better capture fine textures and structural details.

$$\mathcal{L}_{\text{edge}} = \sum \sqrt{(\nabla(R^L) - \nabla(G^L))^2 + \epsilon^2} \tag{5}$$

where $\nabla$ denotes the Laplacian operator, and $R^L$, $G^L$ are the rendered and ground-truth luminance channels, respectively. In addition, we also deploy the $\mathcal{L}_1$ and $\mathcal{L}_{D-SSIM}$ from 3DGS and the overall objective for the $L$ channel can be written as:

$$\mathcal{L}_l = (1-\beta)\mathcal{L}_1 + \beta\mathcal{L}_{D-SSIM} + \mathcal{L}_{edge}, \tag{6}$$

where $\beta$ is set to 0.2 similar to the original 3DGS Kerbl et al. (2023). For the $ab$ channels, which primarily contain low-frequency chromatic information, we exclude the edge loss term and utilize only the following objective:

$$\mathcal{L}_{ab} = (1-\beta)\mathcal{L}_1 + \beta\mathcal{L}_{D-SSIM}. \tag{7}$$

**Warm-Up.** To facilitate more accurate 3D structural and temporal motion modeling, we propose to first optimize the structure and deformation of the scene with only luminance supervision for warm-up and then co-optimize the scene with color constraints. More concretely, as illustrated in Fig. 3(c), during the first half of training iterations in 3DGS or 4DGS, we allocate all three sets of SH coefficients to represent the $L$ channel and render a three-channel luminance image, supervised by the luminance loss $\mathcal{L}_l$. In the latter half of the training, we then reassign two of the SH coefficient sets to represent the $a$ and $b$ chrominance channels for color modeling, and optimize them following the above-presented objectives. This warm-up scheme stabilizes early-stage geometry learning while providing a strong initialization for color modeling, ultimately yielding more robust colorization.

Table 1: Quantitative comparisons on static (DL3DV-140, LLFF, and Mip-NeRF 360) 3D scene colorization benchmarks. The top-performing results are highlighted with color.

| Method | Language | | | | Automatic | | | | Reference | | | |
|---|---|---|---|---|---|---|---|---|---|---|---|---|
| | FID↓ | CLIP Score↑ | ME↓ | TC↓ | FID↓ | Colorful↑ | ME↓ | TC↓ | FID↓ | Ref-LPIPS↓ | ME↓ | TC↓ |
| DL3DV-140 (140 Static 3D Scenes) | | | | | | | | | | | | |
| 3DGS+ImageColorizer | - | - | - | - | 63.56 | 28.15 | 0.146 | 0.038 | - | - | - | - |
| 3DGS+VideoColorizer | - | - | - | - | 77.89 | 22.38 | 0.128 | 0.031 | - | - | - | - |
| **Color3D (Ours)** | - | - | - | - | **37.48** | **32.65** | **0.084** | **0.017** | - | - | - | - |
| LLFF (Static 3D Scenes) | | | | | | | | | | | | |
| GaussianEditor | 136.49 | 0.6346 | 0.093 | 0.017 | - | - | - | - | - | - | - | - |
| GenN2N | - | - | - | - | 52.63 | 25.24 | 0.078 | 0.015 | - | - | - | - |
| Ref-NPR | - | - | - | - | - | - | - | - | 111.93 | 0.7958 | 0.112 | 0.018 |
| ColorNeRF | 138.62 | 0.6388 | 0.089 | 0.018 | 58.26 | 21.78 | 0.062 | 0.012 | 103.68 | 0.7891 | 0.086 | 0.013 |
| 3DGS+ImageColorizer | 87.45 | 0.6415 | 0.178 | 0.035 | 40.25 | **34.07** | 0.084 | 0.017 | 67.48 | 0.7366 | 0.131 | 0.023 |
| 3DGS+VideoColorizer | 118.25 | 0.6397 | 0.092 | 0.017 | 51.37 | 26.68 | 0.069 | 0.015 | 88.19 | 0.7493 | 0.095 | 0.015 |
| **Color3D (Ours)** | **62.84** | **0.6544** | **0.051** | **0.009** | **35.10** | 33.99 | **0.056** | **0.007** | **41.36** | **0.6823** | **0.055** | **0.008** |
| Mip-NeRF 360 (Static 3D Scenes) | | | | | | | | | | | | |
| GaussianEditor | 123.71 | 0.6045 | 0.088 | 0.020 | - | - | - | - | - | - | - | - |
| GenN2N | - | - | - | - | 83.69 | 23.44 | 0.137 | 0.032 | - | - | - | - |
| Ref-NPR | - | - | - | - | - | - | - | - | 146.32 | 0.7928 | 0.143 | 0.026 |
| ColorNeRF | 163.12 | 0.6075 | 0.093 | 0.020 | 87.42 | 18.90 | 0.093 | 0.021 | 121.57 | 0.7852 | 0.106 | 0.021 |
| 3DGS+ImageColorizer | 112.33 | 0.6148 | 0.202 | 0.042 | 62.32 | 29.32 | 0.135 | 0.027 | 86.25 | 0.7521 | 0.178 | 0.032 |
| 3DGS+VideoColorizer | 119.24 | 0.6122 | 0.104 | 0.023 | 82.17 | 23.69 | 0.099 | 0.022 | 92.47 | 0.7582 | 0.122 | 0.024 |
| **Color3D (Ours)** | **68.23** | **0.6246** | **0.058** | **0.012** | **39.03** | **33.36** | **0.082** | **0.016** | **48.62** | **0.7028** | **0.079** | **0.015** |

Table 2: Quantitative comparisons on dynamic (DyNeRF and HyperNeRF) 3D scene colorization benchmarks. The top-performing results are highlighted with color.

| Method | Language | | | | Automatic | | | | Reference | | | |
|---|---|---|---|---|---|---|---|---|---|---|---|---|
| | FID↓ | CLIP Score↑ | ME↓ | TC↓ | FID↓ | Colorful↑ | ME↓ | TC↓ | FID↓ | Ref-LPIPS↓ | ME↓ | TC↓ |
| DyNeRF (Dynamic 3D Scenes) | | | | | | | | | | | | |
| Instruct 4D-to-4D | 112.35 | 0.6082 | 0.062 | 0.011 | - | - | - | - | - | - | - | - |
| 4DGS+ImageColorizer | 89.39 | 0.6159 | 0.124 | 0.025 | 39.17 | 29.45 | 0.080 | 0.018 | 84.65 | 0.7446 | 0.128 | 0.024 |
| 4DGS+VideoColorizer | 88.55 | 0.6199 | 0.071 | 0.014 | 42.36 | 23.42 | 0.073 | 0.014 | 91.22 | 0.7598 | 0.098 | 0.018 |
| **Color3D (Ours)** | **58.62** | **0.6271** | **0.041** | **0.007** | **37.28** | **32.75** | **0.062** | **0.009** | **52.77** | **0.6717** | **0.052** | **0.008** |
| HyperNeRF (Dynamic 3D Scenes) | | | | | | | | | | | | |
| Instruct 4D-to-4D | 118.62 | 0.6053 | 0.065 | 0.012 | - | - | - | - | - | - | - | - |
| 4DGS+ImageColorizer | 105.23 | 0.6175 | 0.132 | 0.029 | 40.32 | 29.66 | 0.090 | 0.017 | 76.28 | 0.7422 | 0.105 | 0.018 |
| 4DGS+VideoColorizer | 110.46 | 0.6128 | 0.095 | 0.017 | 41.85 | 24.68 | 0.084 | 0.015 | 89.33 | 0.7482 | 0.085 | 0.014 |
| **Color3D (Ours)** | **63.28** | **0.6257** | **0.045** | **0.008** | **37.99** | **33.68** | **0.067** | **0.010** | **58.63** | **0.6610** | **0.055** | **0.008** |

## 4 EXPERIMENTS AND ANALYSIS

### 4.1 EXPERIMENTAL SETTINGS

**Implementation details.** All experiments are conducted using the PyTorch framework on NVIDIA RTX A6000 GPUs. We evaluate our method on three static datasets (DL3DV-140 Ling et al. (2024), LLFF Mildenhall et al. (2019) and Mip-NeRF 360 Barron et al. (2022)) and two dynamic datasets (DyNeRF Li et al. (2022) and HyerNeRF Park et al. (2021b)), all of which are converted to monochrome images for colorization. Structure-from-Motion (SfM) is employed to initialize Gaussian points from monochrome inputs in all experiments. Without loss of generality, both our method and competing approaches deploy the same set of image colorization models for fair comparison: ControlColor Liang et al. (2024) for language-guided colorization, DDColor Kang et al. (2023) for automatic colorization, and UniColor Huang et al. (2022) for reference-based colorization. Apart from our proposed Lab Gaussian representation and optimization objectives, all other settings are kept identical to the original 3DGS and 4DGS frameworks. Notably, our entire pipeline incurs only about eight additional minutes for personalized colorizer tuning, which is considered acceptable for chromatic 3D scene reconstruction.

**Metrics.** We adopt CLIP score Radford et al. (2021) for language-guided colorization to measure text-image alignment, Colorful Score Hasler & Suesstrunk (2003) for automatic colorization to assess color vividness, and Ref-LPIPS Zhang et al. (2023) for reference-based colorization to evaluate perceptual similarity. Fréchet Inception Distance (FID) Heusel et al. (2017) is employed as a standard

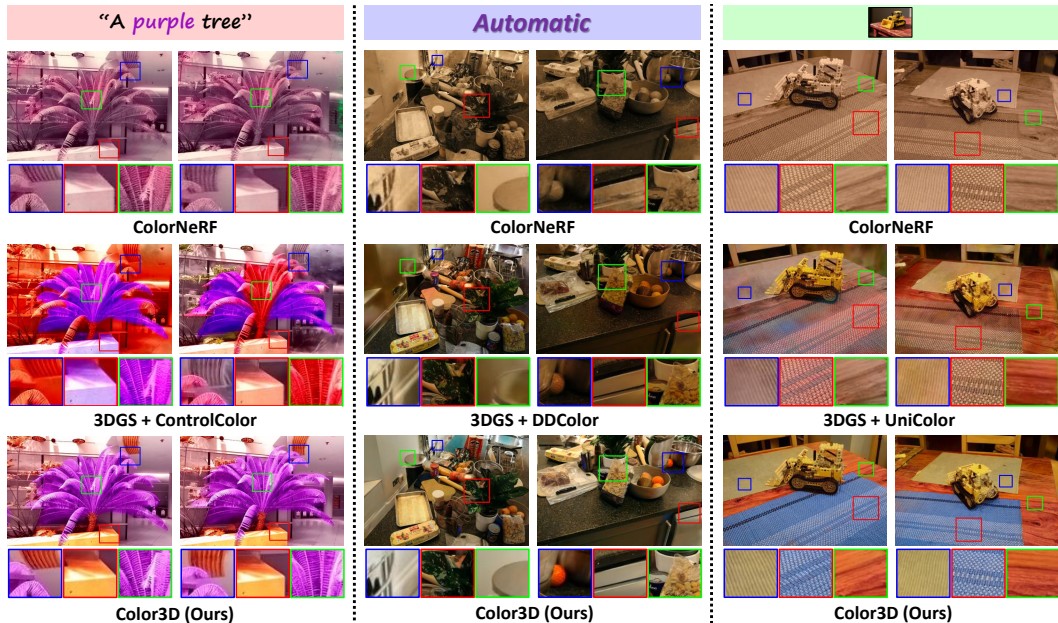

Figure 4: Qualitative comparisons on static 3D scene colorization benchmarks. Our method produces more color-accurate and color-rich results while maintaining multi-view consistency.

metric for assessing the rendering quality of all colorization tasks. Additionally, to evaluate the consistency of colorization across views and time, we propose a metric termed Matching Error (ME). It utilizes a dense image matching model Shen et al. (2024) to identify pixel correspondences between ground-truth color images from different views or time steps, and computes the average color difference at these matched locations in the rendered colorized results. Additionally, we compute consistency metrics Dhiman et al. (2023) based on optical-flow correspondences across views and along temporal sequences to more comprehensively assess color consistency across different methods. We report the average of the short-term and long-term consistency scores, which we denote as TC.

## 4.2 RESULTS ON STATIC 3D SCENE COLORIZATION

The experimental results for quantitative comparisons in static 3D scenes are shown in the middle and upper portions of Tab. 1. Apart from image colorization baselines, we also include a state-of-the-art video colorization model, ColorMNet Yang et al. (2024a), as a reference for more comprehensive evaluations. It is observed that our Color3D delivers remarkable performance gains and outperforms all competitive methods significantly across diverse colorization tasks. A direct combination of 3DGS and image colorization models leads to significantly higher Matching Error (ME), indicating severe multi-view inconsistency. While ColorNeRF Cheng et al. (2024b) improves 3D color consistency by averaging inconsistent 2D colorizations, this strategy inevitably sacrifices color richness and controllability, resulting in noticeable degradation in FID scores and other task-specific evaluation metrics. We also demonstrate visual comparisons in Fig. 4. As suggested, our method effectively accommodates various forms of user control and accurately generates the desired colorized scenes while preserving strong multi-view consistency. In contrast, direct integration of image colorizers introduces obvious view inconsistency and visual artifacts, while ColorNeRF produces overly uniform color tones with limited color diversity and controllability.

## 4.3 RESULTS ON DYNAMIC 3D SCENE COLORIZATION

The quantitative and qualitative experimental results for the dynamic 3D scenarios are exhibited in the bottom portion of Tab. 1 and Fig. 5. It can be observed that our method significantly outperforms the combination of 4DGS and image colorizers, particularly under language-guided and reference-based settings, achieving better FID scores along with substantial reductions in Matching Error (ME) by 0.83 and 0.70, respectively. Moreover, the visual results further demonstrate that our method offers more precise controllability and superior consistency both spatially and temporally.

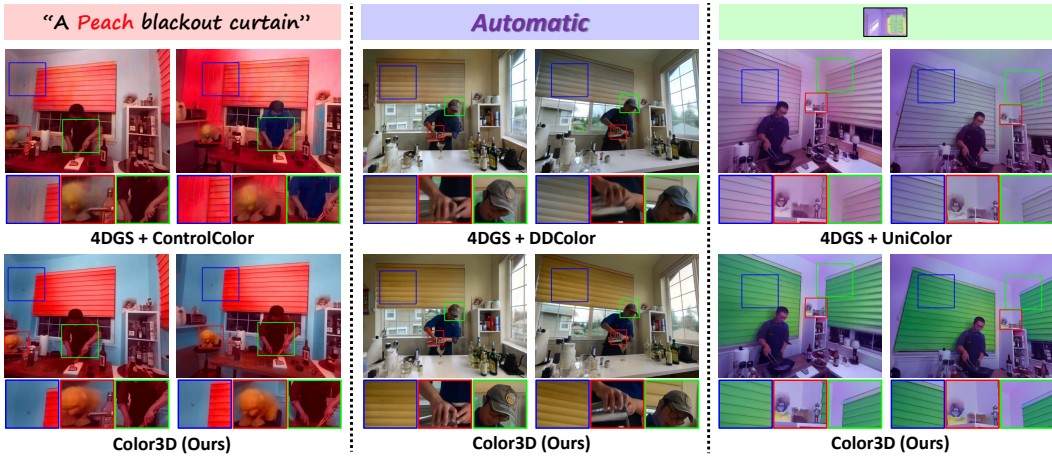

Figure 5: Qualitative comparisons on dynamic 3D scene colorization benchmarks. Our method consistently yields spatial-temporal coherent results with vivid and perceptually realistic color.

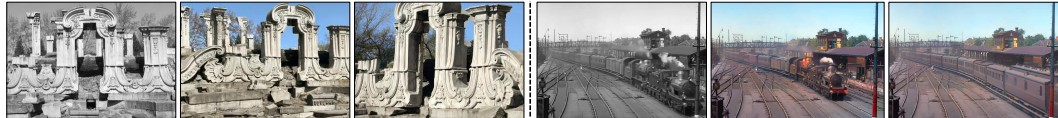

Figure 6: *Left:* Novel views of static 3D scene colorization from in-the-wild monochrome multi-view images. *Right:* Novel views of dynamic 3D scene colorization from a historical monochrome video.

## 4.4 RESULTS ON REAL-WORLD APPLICATIONS

To more rigorously validate the effectiveness of our proposed Color3D, we also conduct experiments on collected in-the-wild monochrome multi-view images and old movies. As illustrated in Fig. 6, our method is capable of producing realistic and vivid colorizations while maintaining impressive color consistency across viewpoints and time, demonstrating the effectiveness of our approach in revitalizing historical or legacy visual content.

## 4.5 DIAGNOSTIC ANALYSIS OF COLOR CONSISTENCY

To better understand the internal behavior of our personalized colorizer and further validate the mechanism behind its strong cross-view consistency, we conducted a diagnostic experiment focusing on the feature representations of the trained model.

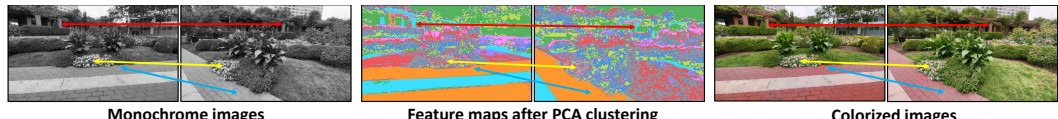

Figure 7: Visualization of cross-view feature consistency and color consistency.

Specifically, we analyze whether the colorizer produces consistent and stable features across different viewpoints of the same 3D scene content — a crucial prerequisite for achieving view-consistent color predictions. To this end, we extract features from two distinct camera viewpoints and project them into a low-dimensional space using PCA for visualization. The reduced features are then clustered and color-coded such that similar features share the same color, enabling intuitive comparison across views. As illustrated in Fig. 7, for the same content in 3D space — such as flower clusters, pillars, or stone paths — the extracted features remain consistent and stable across different viewpoints. This indicates that the trained colorizer maintains highly stable feature embeddings for the same 3D content despite changes in camera perspective. Such feature-level consistency directly aligns with the model's final color outputs and provides strong empirical evidence supporting the design rationale of our personalized colorizer. Please refer to Appendix A.3 for more rigorously theoretical analysis.

## 4.6 ABLATION STUDIES

In Tab. 3, we conduct ablation experiments on the dedicated components introduced in Color3D. The effectiveness of each proposed component in Color3D is evaluated by gradually integrating them into

the model, revealing their individual contributions to the overall performance. More detailed analyses and visual demonstrations are as below.

Table 3: Ablation studies on language-guided colorization on the Mip-NeRF 360 dataset.

| Variant | FID↓ | CLIP Score↑ | ME↓ | TC↓ |
|---|---|---|---|---|
| Baseline | 115.66 | 0.6145 | 0.078 | 0.018 |
| + Key View Selection | 100.28 (+15.38) | 0.6175 (+0.0030) | 0.074 (+0.004) | 0.017 (+0.001) |
| + Single View Augmentation | 85.25 (+15.03) | 0.6196 (+0.0021) | 0.069 (+0.005) | 0.016 (+0.001) |
| + Fine-Tuning-Based Colorizer | 75.48 (+9.77) | 0.6225 (+0.0029) | 0.064 (+0.005) | 0.013 (+0.003) |
| + Lab Gaussian | 68.23 (+7.25) | 0.6246 (+0.0021) | 0.058 (+0.006) | 0.012 (+0.001) |

**Effort of key view selection.** As shown in Fig. 8(a), randomly selected key view often fail to sufficiently capture the full content distribution of the scene, leading to fine-tuned colorizers exhibiting limited color richness when generalized to unseen regions of the scene. In contrast, our key view selection scheme identifies more informative and representative key view, thereby enabling the propagation of richer and more diverse colorizations across novel views.

**Effort of single view augmentation.** Fig. 8(b) demonstrates that the proposed single view augmentation strategy can effectively extrapolate potential visual content beyond the selected view and expand the sample space, thereby facilitating the colorizer to generate more plausible and enriched colorizations for objects not originally visible in the key view and better preserve scene saturation.

**Effort of personalized colorizer.** As illustrated in Fig. 8(c), a randomly initialized colorizer, due to its suboptimal feature extraction capabilities, tends to cause color drifting and miscolorization when generalized to novel views. On the contrary, our personalized colorizer fine-tunes a pre-trained DDColor encoder to learn a high-level semantically aware color mapping, ensuring more accurate and consistent colorization across novel views in the scene.

**Effort of Lab Gaussian.** As depicted in Fig. 8(d), the vanilla RGB Gaussian representation, which entangles luminance and chrominance, tends to introduce blurring artifacts and ghosting since color perturbations will simultaneously affect all three channels. In contrast, our proposed Lab Gaussian representation decouples luminance from predicted chrominance for independent optimization, resulting in significantly sharper textures and more faithful structural details.

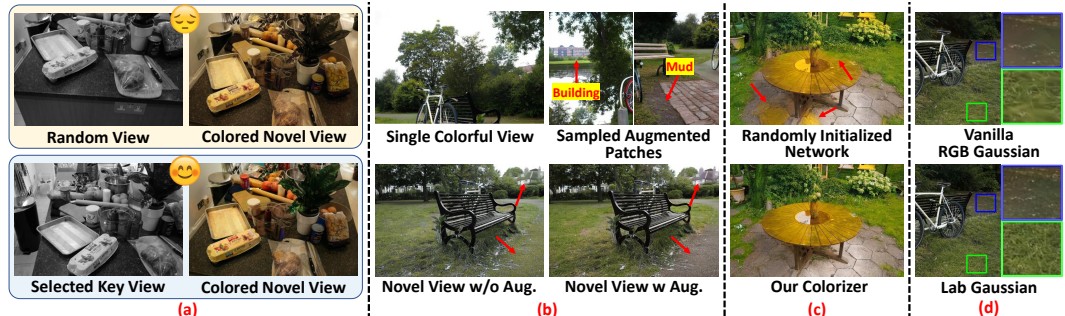

Figure 8: Visual ablation study illustrating the impact of (a) key view selection, (b) single view augmentation, (c) personalized colorizer design, and (d) Lab Gaussian representation.

## 5  CONCLUDING REMARKS

In this work, we propose Color3D, an innovative framework that can efficiently reconstruct colorful static and dynamic 3D scenes from monochromatic inputs with desirable controllability and consistency. In contrast to existing methods that rely on color averaging strategies to enforce multi-view consistency, which often leads to monotonous hues and uncontrollable results, our approach elegantly tackles cross-view and cross-time consistency by fine-tuning a personalized colorizer using a single colorized view. By specializing in the deterministic color mapping underlying this reference view, the colorizer can consistently propagate the intended colors to novel views and frames, preserving color consistency without sacrificing vividness and diversity. Such a design also empowers users to control the overall scene via controlling only one view, enabling flexible and concise controlled colorization. Extensive experiments on various static and dynamic 3D scene colorization benchmarks manifest the effectiveness, superiority, and controllability of our method.

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

## A  APPENDIX

### A.1  3DGS & 4DGS

**3D Gaussian Splatting** (3DGS) is an emerging and highly effective representation for novel view synthesis, which models a 3D scene as a set of $G$ anisotropic Gaussians. Each Gaussian $g_i$ defines a spatial density function:

$$g_i(x) = \exp\left(-\frac{1}{2}(x - \mu_i)^\top \Sigma_i^{-1}(x - \mu_i)\right), \tag{8}$$

where $1 \le i \le G$, $\mu_i \in \mathbb{R}^3$ denotes the mean (i.e., the Gaussian center), and $\Sigma_i \in \mathbb{R}^{3 \times 3}$ is the covariance matrix encoding its anisotropic shape. In practice, $\Sigma_i$ is parameterized by a positive scaling vector $s_i \in \mathbb{R}_+^3$ and a unit quaternion $q_i \in \mathbb{R}^4$ representing its orientation. Each Gaussian is further associated with an opacity value $\alpha_i \in \mathbb{R}_+$ and a set of spherical harmonics (SH) coefficients $c_i$ for view-dependent RGB color modeling.

**4D Gaussian Splatting** (4DGS) extends 3DGS by introducing a learnable deformation field, allowing dynamic scenes to be modeled via a canonical set of 3D Gaussians. Temporal dynamics are captured by predicting time-dependent offsets in position, rotation, and scale, formulated as:

$$g_i = (\mu_i + \delta\mu_i, q_i + \delta q_i, s_i + \delta s_i, \alpha_i, c_i), \quad (\delta\mu_i, \delta q_i, \delta s_i) = \mathcal{F}(\mu_i, t), \tag{9}$$

where $\mathcal{F}$ indicates deformation prediction module.

### A.2  MORE METHOD DETAILS AND ANALYSIS

**Structure details of pensorlized colorizer.** As shown in Fig. 9(a), our personalized colorizer adopts an encoder-decoder structure, augmented with lightweight adapter modules to enable efficient per-scene fine-tuning. The encoder comprises several pre-trained ConvNeXt blocks, each equipped with an inserted adapter. During fine-tuning, the backbone ConvNeXt blocks from pre-trained DDColor Kang et al. (2023) are frozen, and only the adapters and decoder are updated. Each adapter (Fig. 9(b)) follows a bottleneck design consisting of a depth-wise convolution, nonlinearity, and point-wise convolution. Given an input feature map, the adapter first reduces its channel dimension, applies a nonlinear transformation (ReLU), and projects it back to the original size. A residual connection modulated by an attention weight further refines the adaptation. The decoder is composed of a stack of simple and lightweight CNN blocks, each consisting of two convolutional layers, one instance normalization layer, and LeakyReLU serves as activation function. Our structure design effectively allows the colorizer to specialize in learning scene-specific color mappings from given views while maintaining the semantic representation capacity gained from the pre-trained image colorization model.

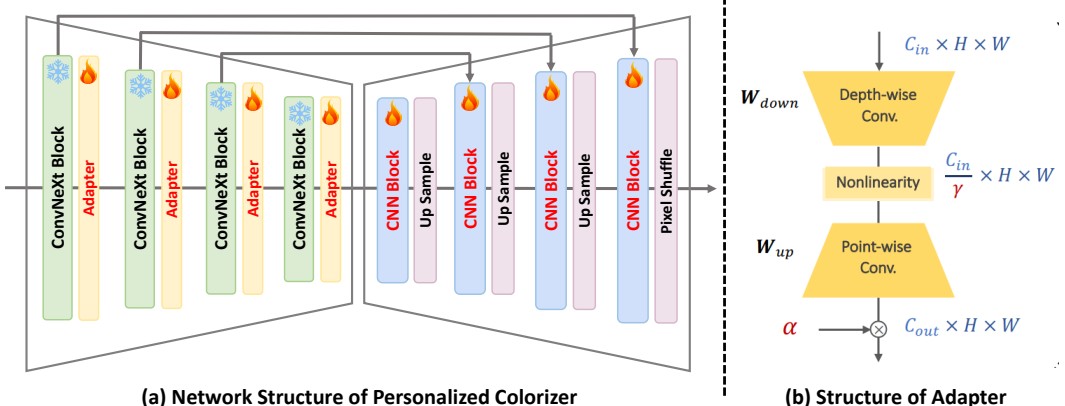

**(a) Network Structure of Personalized Colorizer**  **(b) Structure of Adapter**

Figure 9: (a) Overall architecture of the proposed personalized colorizer. Only the adapters and decoder are updated during fine-tuning. (b) Structure of the adapter module Chen et al. (2024a), which efficiently injects scene-specific colorization capabilities.

**Why can the personalized colorizer achieve consistent color propagation?** Here, we intuitively explain why a colorizer trained on augmented samples from a single colorized view can maintain color consistency when generalized to novel viewpoints and video frames. We illustrate this by comparing a standard image colorization model with our per-scene personalized colorizer.

- **Standard image colorization model.** Typically, training a general-purpose image colorization model requires large-scale luminance-color image pairs. However, the same semantic object may appear in different colors across images, for example, a door might be white in one image but black in another. Consequently, the model learns an uncertain one-to-many color mapping and tend to infer colors based on contextual cues rather than object identity, making it prone to inconsistent color predictions when facing slight viewpoint or motion changes.
- **Our per-scene personalized colorizer.** In contrast, our personalized colorizer is trained on augmented samples derived from a single key view, where the same objects maintain consistent colors across all variations. These augmentations simulate changes in deformation, context, viewpoint, motion, unseen content, etc. This setup encourages the model to learn a scene-specific, variation-agnostic, one-to-one color mapping. Leveraging the inherent inductive bias of deep neural networks Neyshabur et al. (2017); Battaglia et al. (2018), the trained colorizer can predict consistent colors for novel views and video frames by mapping the same content to the corresponding colors it learned during training.

## A.3 THEORETICAL ANALYSIS OF WHY THE LEARNED MAPPING REMAINS CONSISTENT UNDER LARGE VIEW CHANGES.

To help readers gain a deeper understanding of how the proposed personalized colorizer achieves multi-view consistent color propagation, this section provides rigorous convergence guarantees that theoretically establish its effectiveness.

**Notation.** Let $\mathcal{S}$ denote the scene surface (luminance). For a 3D surface point $X \in \mathcal{S}$ and a camera/view $v \in \mathcal{V}$, let $I_v(X)$ denote the image patch (or local crop) containing the projection of $X$. Let $\phi : \mathcal{I} \to \mathbb{R}^d$ be the pretrained encoder (frozen in our pipeline), and $\psi : \mathbb{R}^d \to \mathbb{R}^c$ be the lightweight adapter / color decoder (which we optimize per scene). Define the composite $F := \psi \circ \phi$. For two views $v, v'$ of the same 3D point $X$ we define the color discrepancy

$$\Delta_c(X; v, v') := \|F(I_v(X)) - F(I_{v'}(X))\|.$$

**High-level goal.** Show (under explicit assumptions) (i) a decomposed upper bound on $\Delta_c$ that isolates encoder and adapter/decoder contributions, and (ii) how single-view augmentation + consistency training reduces the bound's dominant terms. Provide (iii) convergence statements for adapter/decoder optimization under clearly stated assumptions.

### A.3.1 ASSUMPTIONS

We explicitly state the assumptions used below. These are empirical/practical but necessary for rigorous statements.

- **A1** (**Encoder local Lipschitzness**) There exists $L_\phi > 0$ and $\epsilon_\phi \geq 0$ such that for any patches $I, I'$,
  $$\|\phi(I) - \phi(I')\| \leq L_\phi \|I - I'\| + \epsilon_\phi.$$
  The term $\epsilon_\phi$ models finite-capacity / non-smooth residual errors and is assumed small when $I, I'$ are semantically similar.
- **A2** (**Approximate viewpoint equivariance**) For typical geometric viewpoint transform $T$ relating views of the same 3D point,
  $$\|\phi(I) - \phi(T[I])\| \leq \delta_{\mathrm{eq}},$$
  where $\delta_{\mathrm{eq}}$ is small under moderate viewpoint changes (this expresses encoder's empirical multi-view invariance).
- **A3** (**Adapter local Lipschitzness**) The adapter $\psi$ is locally Lipschitz on the relevant feature domain: there exists $L_\psi > 0$ such that for features $z, z'$,
  $$\|\psi(z) - \psi(z')\| \leq L_\psi \|z - z'\|.$$
  In practice, $L_\psi$ can be controlled via architecture size and weight regularization.

**A4** (**Augmentation consistency objective**) Training uses an empirical loss combining supervised reference fit (the augmented images) and a consistency term:

$$\mathcal{L}(\theta) \;=\; \mathbb{E}_X \Big[ \sum_k \ell\big(\psi_\theta(\phi(T_k[I(X)])), c^\star(X)\big) \Big] + \lambda \mathbb{E}_X \Big[ \sum_{k \neq k'} \|\psi_\theta(\phi(T_k[I])) - \psi_\theta(\phi(T_{k'}[I]))\|^2 \Big],$$

where $\theta$ are adapter parameters, $T_k$ are single-view augmentations, and $\lambda \geq 0$.

### A.3.2  LEMMA 1 (FEATURE DIFFERENCE UPPER BOUND).

Under A1–A2, for any two views $v, v'$ of the same 3D point $X$,

$$\|\phi(I_v(X)) - \phi(I_{v'}(X))\| \leq L_\phi \|I_v(X) - I_{v'}(X)\| + \epsilon_\phi + \delta_{\mathrm{eq}}.$$

*Proof.* By triangle inequality, write $I_{v'} \approx T[I_v] + \eta$ where $\eta$ models sampling/noise/occlusion residual. Then

$$\|\phi(I_v) - \phi(I_{v'})\| \leq \|\phi(I_v) - \phi(T[I_v])\| + \|\phi(T[I_v]) - \phi(I_{v'})\|.$$

By A2 the first term is $\leq \delta_{\mathrm{eq}}$. For the second term apply A1 with $I' = I_{v'}$ and $I = T[I_v]$, noting $\|T[I_v] - I_{v'}\| \leq \|I_v - I_{v'}\| + \|\eta\|$; absorbing $\|\eta\|$ into $\epsilon_\phi$ yields the stated bound. $\qquad\square$

### A.3.3  LEMMA 2 (COLOR DIFFERENCE VIA FEATURE DIFFERENCE).

Under A3, for any two views $v, v'$,

$$\Delta_c(X; v, v') \leq L_\psi \cdot \|\phi(I_v(X)) - \phi(I_{v'}(X))\|.$$

*Proof.* Immediate from the Lipschitz property of $\psi$. $\qquad\square$

### A.3.4  PROPOSITION 1 (DECOMPOSED COLOR DISCREPANCY BOUND).

Combining Lemmas A.3.2 and A.3.3, for any 3D point $X$ and views $v, v'$,

$$\boxed{\Delta_c(X; v, v') \leq L_\psi \Big( L_\phi \|I_v - I_{v'}\| + \epsilon_\phi + \delta_{\mathrm{eq}} \Big).}$$

*Proof.* Direct composition of the two lemmas. $\qquad\square$

**Interpretation.** The bound decomposes color difference into: (i) image-level discrepancy $\|I_v - I_{v'}\|$ (sampling, occlusion), (ii) encoder approximation residual $\epsilon_\phi$, and (iii) encoder equivariance error $\delta_{\mathrm{eq}}$. Reducing $L_\psi$ (smaller adapter sensitivity) or reducing the effective $\|I_v - I_{v'}\|$-sensitivity (via consistency training that shrinks feature/ color variance under $T_k$) will reduce $\Delta_c$.

### A.3.5  HOW CONSISTENCY TRAINING COMPRESSES THE BOUND (INFORMAL BUT QUANTITATIVE SKETCH).

We formalize the effect of the consistency term by considering the second moment (variance) of color outputs across augmentations for a fixed 3D point:

$$V_\psi(X) := \mathbb{E}_T\big[\|\psi(\phi(T[I(X)])) - m_\psi(X)\|^2\big], \qquad m_\psi(X) := \mathbb{E}_T[\psi(\phi(T[I(X)]))].$$

The consistency penalty in $\mathcal{L}$ directly bounds $V_\psi(X)$. Under standard first-order optimality conditions (or gradient flow dynamics), increasing $\lambda$ reduces the equilibrium value of $V_\psi(X)$. Using the Lipschitz relation in Proposition 1 and Jensen, an upper bound on expected squared color discrepancy across sampled view pairs $(v, v')$ can be written as

$$\mathbb{E}_{X,v,v'}[\Delta_c^2] \leq (L_\psi L_\phi)^2 \mathbb{E}[\|I_v - I_{v'}\|^2] + (L_\psi)^2 (\epsilon_\phi + \delta_{\mathrm{eq}})^2.$$

The consistency term reduces the first RHS term by effectively shrinking the empirical $\mathbb{E}[\|I_v - I_{v'}\|^2]$-sensitivity through learned invariance in $\psi$. We denote this shrinkage by a factor $\eta(\lambda) \in (0, 1]$ which empirically decreases as $\lambda$ increases. Thus

$$\mathbb{E}[\Delta_c^2] \leq \eta(\lambda) (L_\psi L_\phi)^2 \mathbb{E}[\|I_v - I_{v'}\|^2] + (L_\psi)^2 (\epsilon_\phi + \delta_{\mathrm{eq}})^2.$$

A precise closed-form for $\eta(\lambda)$ depends on dynamics/architecture; we treat $\eta(\lambda)$ as observable and empirically verifiable (see diagnostics below).

### A.3.6 JACOBIAN / SENSITIVITY PERSPECTIVE (FORMALIZATION).

Let $J_F(I) := \nabla_I F(I)$ denote the Jacobian of $F = \psi \circ \phi$ wrt image pixels (vectorized). By first-order Taylor expansion, for small image perturbations $\delta I$,

$$\|F(I + \delta I) - F(I)\| \leq \|J_F(I)\| \cdot \|\delta I\|.$$

By chain rule $J_F(I) = J_\psi(\phi(I))J_\phi(I)$ and hence $\|J_F\| \leq \|J_\psi\| \cdot \|J_\phi\|$. Consistency training implicitly penalizes $\|J_\psi\|$ (or its expected norm) because reducing output variance across known transforms forces smaller directional derivatives along transform-induced directions; therefore the product $\|J_F\|$ is reduced, decreasing sensitivity to viewpoint-induced perturbations.

### A.3.7 CONVERGENCE RESULTS FOR ADAPTER/DECODER OPTIMIZATION

We give convergence statements: Assume $\psi_\theta$ is $L$-smooth in parameters (i.e., $\nabla_\theta \mathcal{L}$ is Lipschitz with constant $L$) and $\mathcal{L}$ is lower bounded. Then gradient descent (or stochastic gradient descent under standard assumptions) converges to a first-order stationary point: $\lim_{t \to \infty} \|\nabla_\theta \mathcal{L}(\theta_t)\| = 0$. If we further assume the Polyak–Łojasiewicz (PL) condition holds locally (i.e., there exists $\mu > 0$ with $\frac{1}{2}\|\nabla \mathcal{L}(\theta)\|^2 \geq \mu(\mathcal{L}(\theta) - \mathcal{L}^\star)$), then gradient descent converges linearly to a global minimizer with rate $1 - \alpha\mu$.

**Concluding statement.** The preceding derivation has provided (i) a decomposed upper bound on cross-view color discrepancy, (ii) a Jacobian-based mechanism describing how consistency training reduces sensitivity to viewpoint-induced perturbations, and (iii) standard stationary-point results for smooth nonconvex adapter/decoder. The above analysis theoretically substantiates that the proposed personalized colorizer, when optimized with consistency constraints, is guaranteed to produce view-invariant color predictions across varying viewpoints.

## A.4 METHOD EXTENSION

**All-in-one training scheme for large-scale datasets.** Essentially, our personalized colorizer only includes a scene-specific adapter and a lightweight decoder, while the large encoder is shared across all scenes. This design enables us to jointly train a large number of scenes using a routing-based strategy. The semantic illustration of this mechanism is provided in Fig. 10. Concretely, images from different scenes are stacked along the batch dimension, and each sample is routed through its corresponding adapter and decoder based on the scene ID, which enables efficient all-in-one training. By applying this all-in-one personalized colorizer training scheme, we are able to train colorizers for all 140 scenes in the DL3DV-140 dataset within only 55 minutes. This experiment further demonstrates the practicality and efficiency of our approach for large-scale real-world deployment.

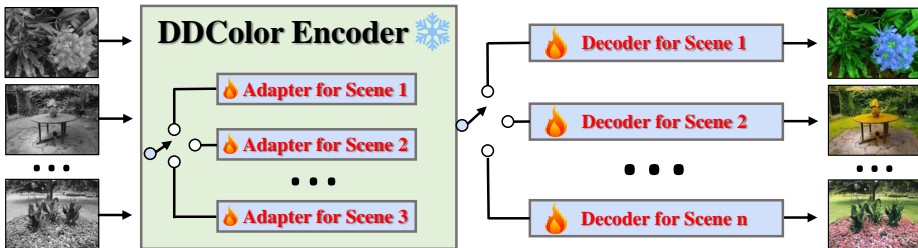

Figure 10: Illustration of the proposed all-in-one training scheme for large-scale datasets.

**Generalized to feed-forward style paradigm.** To further explore the applicability of the proposed framework beyond per-scene optimization, we investigate its extension to a feed-forward 3D generation paradigm. As illustrated in Fig. 11, we first apply our proposed single-view generative augmentation strategy on a large-scale image dataset to synthesize substantial spatial and viewpoint variations. Using only one colored image as the key view, we train a generalizable personalized colorizer based on a transformer network that learns to colorize all augmented views in a feed-forward manner. At inference time, we follow the exact pipeline described in the paper: we colorize a single input view and then propagate its chroma to other viewpoints using the generalizable personalized colorizer, without any per-scene optimization. The resulting set of multi-view colorized images can

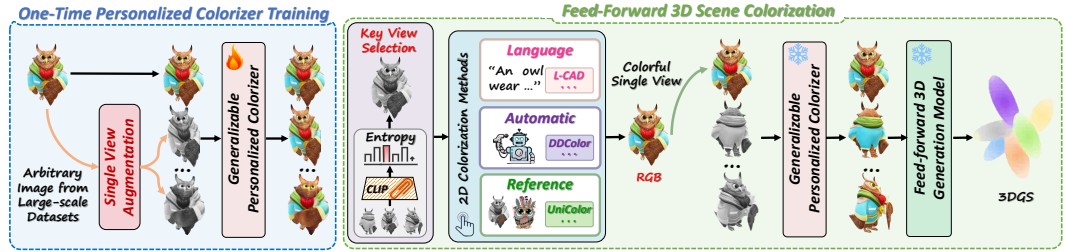

Figure 11: Illustration of the proposed feed-forward style 3D colorization framework.

then be fed into a feed-forward 3D reconstruction model—such as AnySplat Jiang et al. (2025)—to directly obtain a fully colored Gaussian representation. We trained the generalizable personalized colorizer on the COCO dataset and validated the feed-forward 3D colorization pipeline on the DL3DV-10-Benchmark, demonstrating the feasibility of this extension.

As shown in Tab. 4 and Fig. 12, our experiments indicate that the proposed framework already exhibits promising feed-forward 3D colorization potential. While a fully feed-forward solution is indeed appealing, our observations show that a globally generalized colorizer still falls short of our per-scene adapter in both chromatic richness and multi-view consistency, and it suffers from significant resolution degradation. Given the large-baseline viewpoints and high-resolution reconstruction settings considered in our work, the per-scene optimization route remains the most practical and reliable choice at present. Nonetheless, this preliminary investigation highlights a viable direction for future research.

Table 4: Quantitative comparisons of automatic colorization on the DL3DV-140 dataset.

| Method | FID↓ | Colorful↑ | ME↓ | TC↓ |
|---|---|---|---|---|
| Generalizable Colorizer + AnySplat | 82.36 | 24.37 | 0.142 | 0.030 |
| Per-Scene Colorizer + AnySplat | 66.78 | 29.39 | 0.109 | 0.025 |

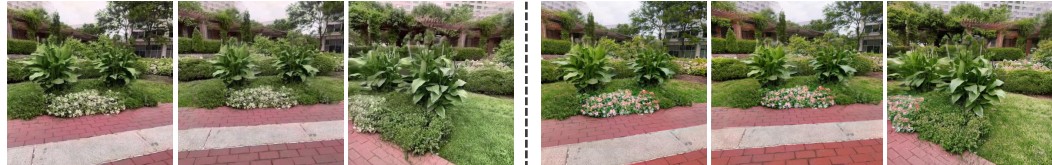

Generalizable Personalized Colorizer + AnySplat · Per-Scene Personalized Colorizer + AnySplat

Figure 12: Visual comparison of generalizable personalized colorizer and per-scene personalized colorizer for feed-forward 3D colorization.

## A.5 MORE EXPERIMENTS

**Dataset details.**

*DL3DV-10-Benchmark* Ling et al. (2024) consists of 140 large-scale 360° real-world scenes. All images are processed at a resolution of $960 \times 540$ for our experiments.

*LLFF Dataset* Mildenhall et al. (2019) consists of 8 forward-facing real-world scenes. Following prior work, we select every eighth image for testing and use the remaining images for training. All images are processed at a resolution of $1008 \times 756$ for our experiments.

*Mip-NeRF 360 Dataset* Barron et al. (2022) comprises 9 real-world scenes, including 5 outdoor and 4 indoor environments. Following the same protocol as LLFF, we use one-eighth of the views for testing and the remaining views for training. All experiments are conducted on images downsampled by a factor of 4 (approximately 1K resolution).

*DyNeRF Dataset* Li et al. (2022) includes six 10-second video sequences captured at 30 fps by 15 to 20 cameras with face forward perspective, involving extended periods and intricate camera motions.

*HyperNeRF* Park et al. (2021b) is another dynamic dataset captured using one or two cameras following relatively straightforward camera trajectories, but with more intricate camera motions.

**More implementation details of pensorlized colorizer tuning.** For the single view augmentation strategy, we first generate nine augmented samples based on the original view. Specifically, four samples are produced using the outpainting model Stable Diffusion Rombach et al. (2022), one sample is extracted from the final frame generated by the image-to-video model Stable Video Diffusion Blattmann et al. (2023), and three samples are obtained by sparsely sampling along an orbit using the novel view synthesis model Stable Virtual Camera Zhou et al. (2025), together with the original view itself. The nine samples were then randomized to apply traditional augmentation operations. During training, the samples input to the personalized colorizer are randomly cropped to $320 \times 320$. We train the personalized colorizer for 4K iterations using the Adam optimizer. During training, only one sample is used per iteration, i.e., $batchsize = 1$, which allows the model to focus on capturing the most distinctive and rich color information from the given sample. The learning rate is initialized to $1 \times 10^4$, which is steadily decreased to $1 \times 10^6$ using the cosine annealing strategy.

**Quantitative comparison with ChromaDistill.** Additionally, following ChromaDistill's experimental setup, we conduct automatic image colorization experiments using BigColor Kim et al. (2022) across four scenes: Cake, Pasta, Buddha, and Leaves. The experimental results are depicted in Tab. 5. It is observed that our method achieves substantially higher consistency than ChromaDistill in both short-term and long-term consistency. Notably, on the Cake scene, our approach improves long-term consistency by 0.023, further demonstrating the superiority of the proposed personalized colorizer in preserving cross-view coherence and validating the effectiveness of our overall framework.

Table 5: Quantitative comparisons with ChromaDistill on cross-view short-term and long-term consistency.

| Method | Short-Term Consistency ↓ | | | | Long-Term Consistency ↓ | | | |
|---|---|---|---|---|---|---|---|---|
| | Cake | Pasta | Buddha | Leaves | Cake | Pasta | Buddha | Leaves |
| ChromaDistill (BigColor) | 0.019 | 0.015 | 0.015 | 0.008 | 0.033 | 0.025 | 0.023 | 0.013 |
| **Color3D (BigColor)** | **0.008** | **0.009** | **0.009** | **0.007** | **0.010** | **0.018** | **0.012** | **0.011** |

**Single view or more?** Due to the inherent instability of 2D colorization models, even small perturbations in the input image can lead to noticeably different color predictions for both individual objects and the overall scene style. When two or more key views are used for training the personalized colorizer, these inconsistencies across views introduce conflicting supervision, which often causes the model to converge to color averaging or to produce view-dependent color shifts during 3D colorization. As shown in Fig. 13, the results obtained by training with two views clearly exhibit such inconsistencies, making it difficult for the personalized colorizer to maintain coherent colors across views. In addition, the quantitative comparison in Tab. 6 shows that using a single key view achieves better consistency and avoids both color drifting and color averaging, leading to superior FID and Colorful scores.

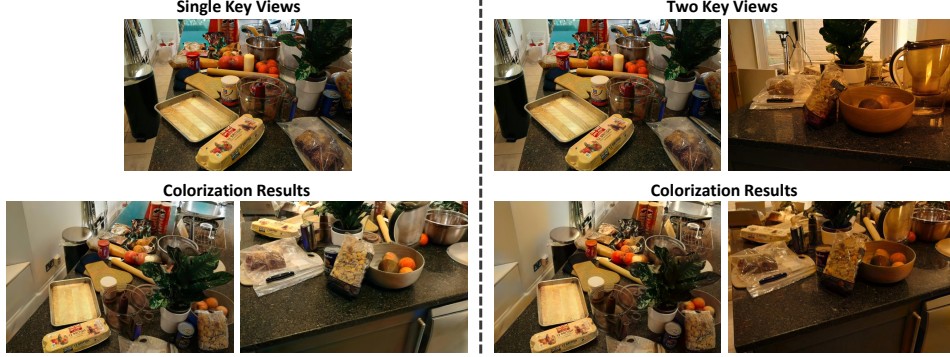

Figure 13: Visual comparison of personalized colorizer training with single vs. two key views.

**Robustness to key view selection.** Our framework demonstrates strong robustness with respect to the choice of the key view. As illustrated in Fig. 14, varying the selected key view consistently leads to coherent and chromatically vibrant 3D colorization results. Even when the first key view contains almost no grass, our approach can still produce reasonable and vibrant colors. Moreover, for distant buildings that do not appear in the key view at all, our method can still infer plausible colors for them.

Table 6: Quantitative comparison of using different numbers of key views for automatic colorization on the Mip-NeRF-360 dataset.

| Number of Key Views | FID↓ | Colorful↑ | ME↓ | TC↓ |
|---|---|---|---|---|
| Three Views | 52.38 | 25.26 | 0.098 | 0.022 |
| Two Views | 45.62 | 27.36 | 0.092 | 0.018 |
| **One View** | **39.03** | **33.36** | **0.082** | **0.016** |

These can be primarily attributed to the generative data augmentation strategy and the colorizer's ability to interpolate colors for unseen content, which together enable stable colorization even when the selected key view is not ideal.

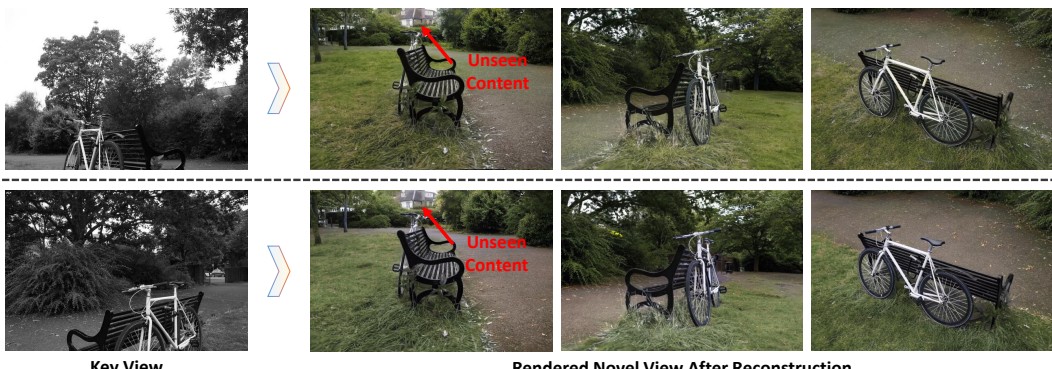

Key View       Rendered Novel View After Reconstruction

Figure 14: Visual comparison of the colorization results with different key views.

**What is the color of semantically similar but visually different objects?** To further evaluate the generalization capability of our personalized colorizer, we conduct an experiment on scenes containing semantically similar yet visually distinct objects. As shown in Fig. 15, our method consistently assigns plausible and stable colors even when object appearances in the target views deviate from those seen in the key view. This behavior arises from two key design choices. First, the frozen encoder, pre-trained on large-scale datasets, provides robust and viewpoint-invariant visual features. During optimization, the encoder yields consistent representations for the same object across different viewpoints while still capturing feature differences for objects that are semantically similar but visually distinct. Second, the decoder, trained jointly on the key view and our generatively augmented images, learns a broader color mapping space. This enables it to perform effective color interpolation when encountering unseen visual features.

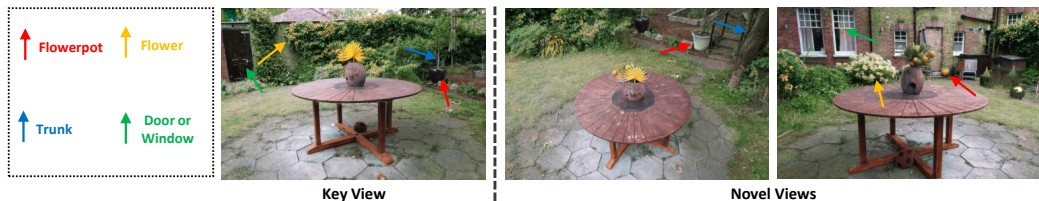

Key View       Novel Views

Figure 15: Visual comparison of the colorization results on objects that share semantic similarity but differ in visual appearance. More views can be found in Fig. 23

**Performance under challenging lighting conditions.** We also demonstrate the performance of our method under challenging lighting conditions in Fig. 16. Specifically, we evaluate scenes with complex lighting, including lamps, reflections, shadows, and highly specular materials. Our results show that the proposed method effectively handles these challenging scenarios and produces consistent and reasonable colorization even in regions affected by shadows and projections. These findings further validate the reliability and effectiveness of our approach in practical applications.

**Optimization efficiency.** In Tab. 7, we compare the optimization time across different methods. While our Stage 1 introduces an extra 8 minutes, the Stage 2 optimization is significantly faster than that of 3DGS+ImageColorizer, and the overall time is even shorter than reconstructing a 3D scene from normal colored images using 3DGS. This efficiency is mainly due to our proposed

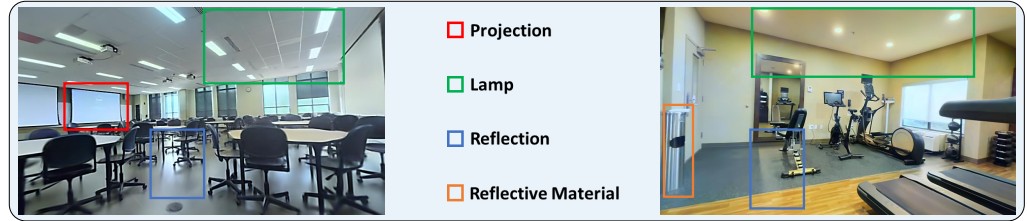

Figure 16: Visual results of our method under challenging lighting conditions.

Lab-space optimization strategy, which first optimizes the L channel to capture structural information and then learns color modeling. This strategy substantially reduces redundant Gaussian clones and splits, allowing fewer Gaussian primitives to model the scene more effectively. In contrast, 3DGS+ImageColorizer splits a large number of Gaussian primitives to model multi-view inconsistent colors, resulting in longer optimization times than standard 3DGS reconstruction. ColorNeRF requires an even longer 16 hours to reconstruct a single scene. These results demonstrate that our method is more efficient than baseline reconstruction approaches, and that per-scene optimization is acceptable in practice.

Table 7: Average optimization time on Mip-NeRF-360 (static) and DyNeRF (dynamic) datasets.

| Dataset | Ours | | 3D/4DGS | | ColorNeRF |
|---|---|---|---|---|---|
| | Stage 1 | Stage 2 | + ImageColorizer | Original | |
| Static (Mip-NeRF-360) | 8 mins | 31 mins | 50 mins | 42 mins | 16 h |
| Dynamic (DyNeRF) | 8 mins | 38 mins | 54 mins | 47 mins | – |

**Generalization on dissimilar scenes.** As discussed in Section 4.5 and illustrated in Fig. 7, our personalized colorizer can be transferred to similar scenes without retraining, which can reduce the optimization burden for large-scale similar scenes. Here, we further evaluate the performance of the proposed method on more challenging, substantially different scenes. Specifically, we train the colorizer on the "Garden" scene from Mip-NeRF 360 and transfer it to the "Room" scene. As shown in Fig. 17, our method achieves suboptimal performance when the target scene is very different from the training scene. For example, some sofa regions inherit green tones from the source scene, and the overall color richness is reduced. Nevertheless, the colorizer still produces overall consistent and stable results and, in particular, performs reasonably well for elements that share similar semantic properties, such as the vegetation outside the window. In summary, the personalized colorizer demonstrates potential generalization to similar scenes, reducing the need for retraining across many visually similar scenes in large-scale datasets. However, we still advocate scene-specific fine-tuning to achieve optimal 3D colorization results, which remains our recommended practice in this work.

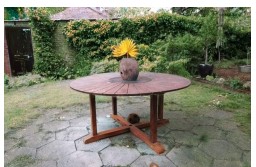 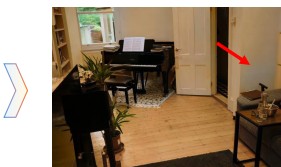 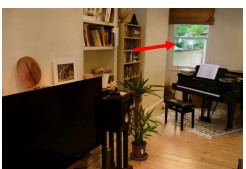 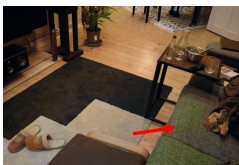

Figure 17: Visualization of the generalization performance on dissimilar scene.

**Image and video colorizers vs. per-scene colorizer.** We evaluate the effectiveness of our per-scene colorizer design by comparing it with two representative baselines: the image colorization model DDColor Kang et al. (2023) and the video colorization method ColorMNet Yang et al. (2024a), as illustrated in Fig. 18. It can be observed that DDColor suffers from severe cross-view color inconsistencies due to its per-image one-to-many color mapping nature. Treating multi-view images as a video sequence and applying video colorization similarly fails to resolve this issue, since the viewpoint changes inherent in 3D capture far exceed the temporal variations commonly seen in video data. In contrast, our personalized colorizer learns a scene-specific, one-to-one color mapping from the reference view and leverages its inherent inductive bias to consistently project colors to

corresponding content in novel views and time steps. This design effectively addresses the challenge of color consistency in both static and dynamic 3D scenes.

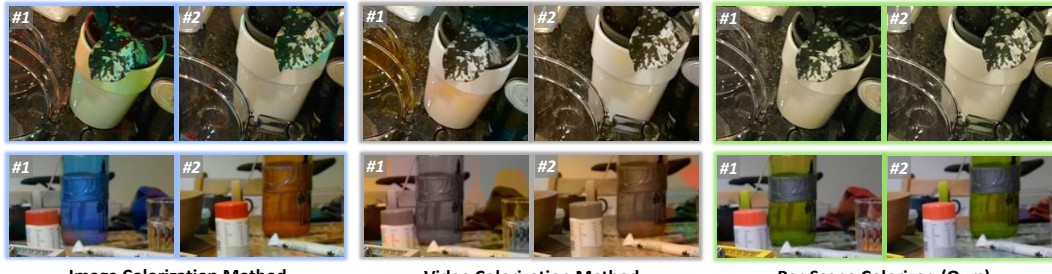

Figure 18: Visual ablation study illustrating the effectiveness of the per-scene colorizer design.

**Effort of 3D representations.** We further investigate how different 3D representations impact colorization performance. As shown in Tab.8, compared to 3DGS, combining NeRFMildenhall et al. (2021) or NeRFPlayer Song et al. (2023) with image colorizers yields suboptimal consistency, reflected by an increase of 0.023 in ME. This is mainly because the explicit modeling of 3DGS gives it an inherent resistance and geometric priors, which can naturally alleviate color inconsistency to some extent. In addition, we replace the Lab Gaussian representation in Color3D with NeRF and NeRFPlayer, respectively. The results underscore the effectiveness of our proposed Lab Gaussian representation while also demonstrating that the Color3D framework can be adapted to various 3D reconstruction backbones with reasonable performance.

Table 8: Quantitative comparisons of language-guided 3D scene colorization on Mip-NeRF 360 and DyNeRF datasets.

| Method | Language-Guided Colorization | | | |
|---|---|---|---|---|
| | FID↓ | CLIP Score↑ | ME↓ | TC↓ |
| Mip-NeRF 360 (Static 3D Scenes) | | | | |
| NeRF+ImageColorizer | 104.85 | 0.6153 | 0.225 | 0.046 |
| 3DGS+ImageColorizer | 112.33 | 0.6148 | 0.202 | 0.042 |
| Color3D-NeRF | 76.68 | 0.6213 | 0.065 | 0.014 |
| **Color3D (Ours)** | **68.23** | **0.6246** | **0.058** | **0.012** |
| DyNeRF (Dynamic 3D Scenes) | | | | |
| NeRFPlayer+ImageColorizer | 86.77 | 0.6148 | 0.133 | 0.027 |
| 4DGS+ImageColorizer | 89.39 | 0.6159 | 0.124 | 0.025 |
| Color3D-NeRFPlayer | 63.29 | 0.6243 | 0.046 | 0.009 |
| **Color3D (Ours)** | **58.62** | **0.6271** | **0.041** | **0.008** |

**Can editing or stylization methods work for colorization?** In Tab. 1 of the main text, we present quantitative comparisons with a representative 3D editing method (GaussianEditor Chen et al. (2024b)) and a 3D stylization method (Ref-NPR Zhang et al. (2023)), both of which yield suboptimal performance on colorization tasks. To demonstrate their visual limitations, we further provide qualitative comparisons in Fig. 19.

It is observed that GaussianEditor struggles to comply with color editing instructions when applied to monochromatic radiation fields, primarily prioritizing semantic fidelity over chromatic accuracy. As a result, the edited outputs, although containing colors, often lack chromatic realism and diversity, and may also suffer from structural artifacts. Similarly, Ref-NPR primarily focuses on transferring global appearance by relying on perceptual constraints, but fails to accurately capture local color values. In contrast, colorization aims to produce plausible, detailed, and spatially coherent colors that align with the underlying scene content. Consequently, editing and stylization methods are inherently ill-suited for faithful 3D scene colorization. Moreover, their controllability remains limited: editing is typically driven by text prompts, while stylization relies on reference images to guide appearance transfer.

**Universal color manipulation framework.** Beyond colorizing 3D scenes from monochromatic inputs, our proposed scheme can also be naturally extended to 3D scene recoloring. Specifically, we apply InstructPix2Pix Brooks et al. (2023) to edit the colors of a single key view and leverage SAM

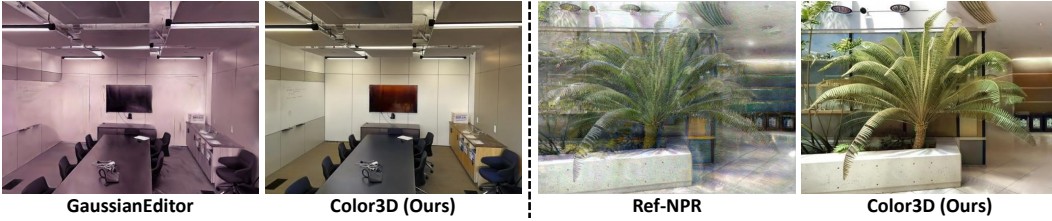

| GaussianEditor | Color3D (Ours) | Ref-NPR | Color3D (Ours) |

Figure 19: Visual comparisons with the 3D editing method GaussianEditor Chen et al. (2024b), and the 3D stylization method Ref-NPR Zhang et al. (2023).

Kirillov et al. (2023) to constrain the editing within specified regions. For the personalized colorizer, the input is the original image, and the output is its recolored version. As illustrated in Fig. 20, our method not only enables colorization from monochromatic inputs, but also supports high-fidelity and consistent color manipulations, demonstrating its robustness and versatility.

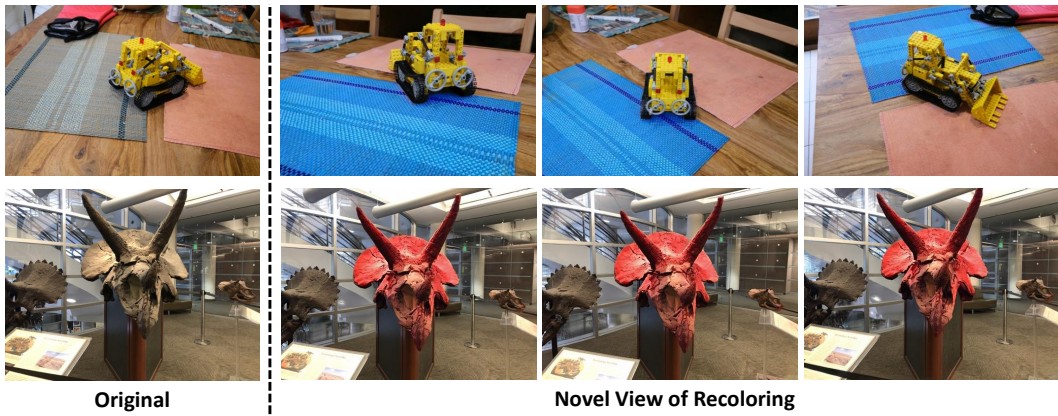

| Original | Novel View of Recoloring |

Figure 20: Visual results of 3D recoloring.

**User study.** To further assess the perceptual quality of the generated results, we conduct a comprehensive user study, acknowledging that subjective human perception remains a critical benchmark in the field of colorization. Specifically, we design 5 evaluation questions covering multiple aspects, including color richness, consistency, aesthetic preference, image quality, and the fidelity of alignment between the synthesized views and the provided conditions (text, exemplars). Participants are asked to rate each sample on a scale of 0 to 5 for each aspect. For a fair comparison, we recruit 30 participants, each evaluating 10 sets of comparisons across different methods. Aggregated results are shown in Fig. 21. The naive combination of 3DGS with off-the-shelf image colorizers yields reasonably rich colors but suffers from significant inconsistency and reduced visual quality. While ColorNeRF demonstrates better consistency, it often lacks color diversity and visual appeal. In contrast, our method consistently achieves higher user ratings across all aspects, highlighting its advantage in terms of perceptual quality and user preference.

**More visual results.** We further present additional colorization renderings of our Color3D in Fig. 22, Fig. 23, and Fig. 24. Our approach exhibits strong consistency across both spatial and temporal dimensions, delivering vibrant and semantically coherent colorizations that faithfully adhere to user-specified intent.

## A.6 LIMITATIONS AND FUTURE WORK

One potential limitation, shared with other single-view methods, is that our approach tends to favor assigning plausible colors consistent with the observed input rather than hallucinating entirely novel chromatic content when confronted with highly out-of-domain views (e.g., reconstructing a large room with multiple unseen bedrooms). In future work, we plan to further explore the integration of generative priors to enrich color diversity under drastically unseen viewpoints while maintaining color consistency. Moreover, the per-scene personalization strategy introduced in this work is highly general and carries substantial potential beyond colorization. Extending this paradigm to

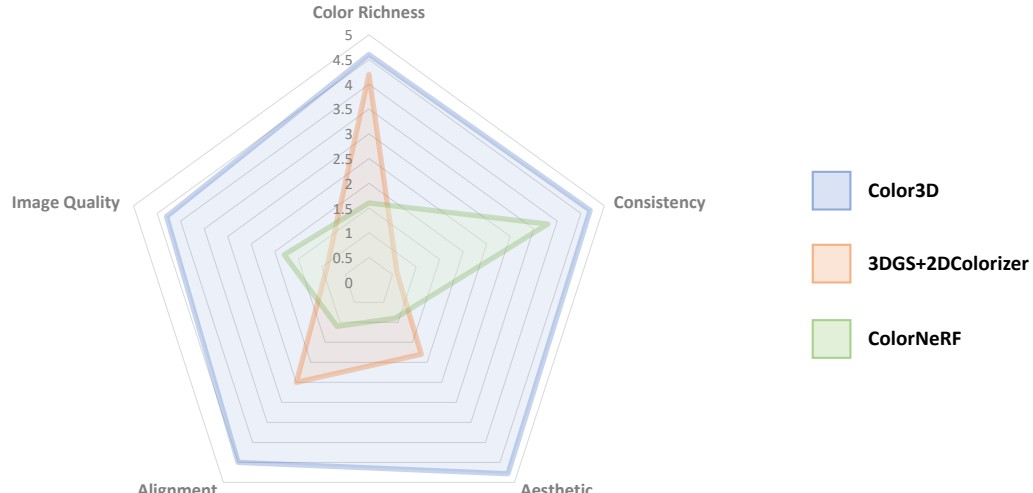

Figure 21: User study results. Our method demonstrates superior performance across all evaluated aspects.

other scene attributes—such as illumination enhancement, white balance adjustment, and style transfer—represents a promising avenue toward broader and more versatile scene-level editing capabilities.

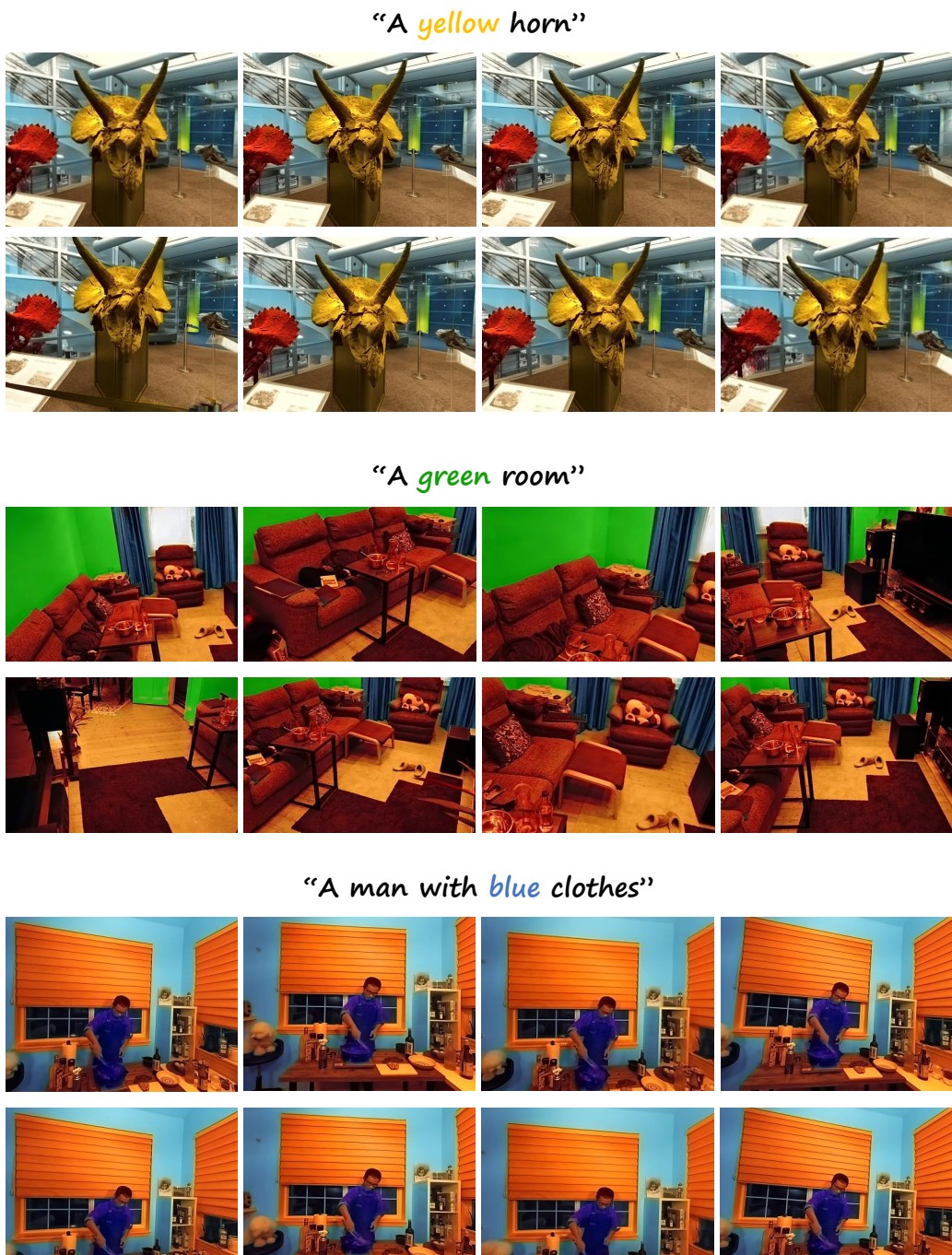

Figure 22: Visual results of language-guided 3D scene colorization. From top to bottom are samples from the LLFF, Mip-NeRF 360, and DyNeRF datasets.

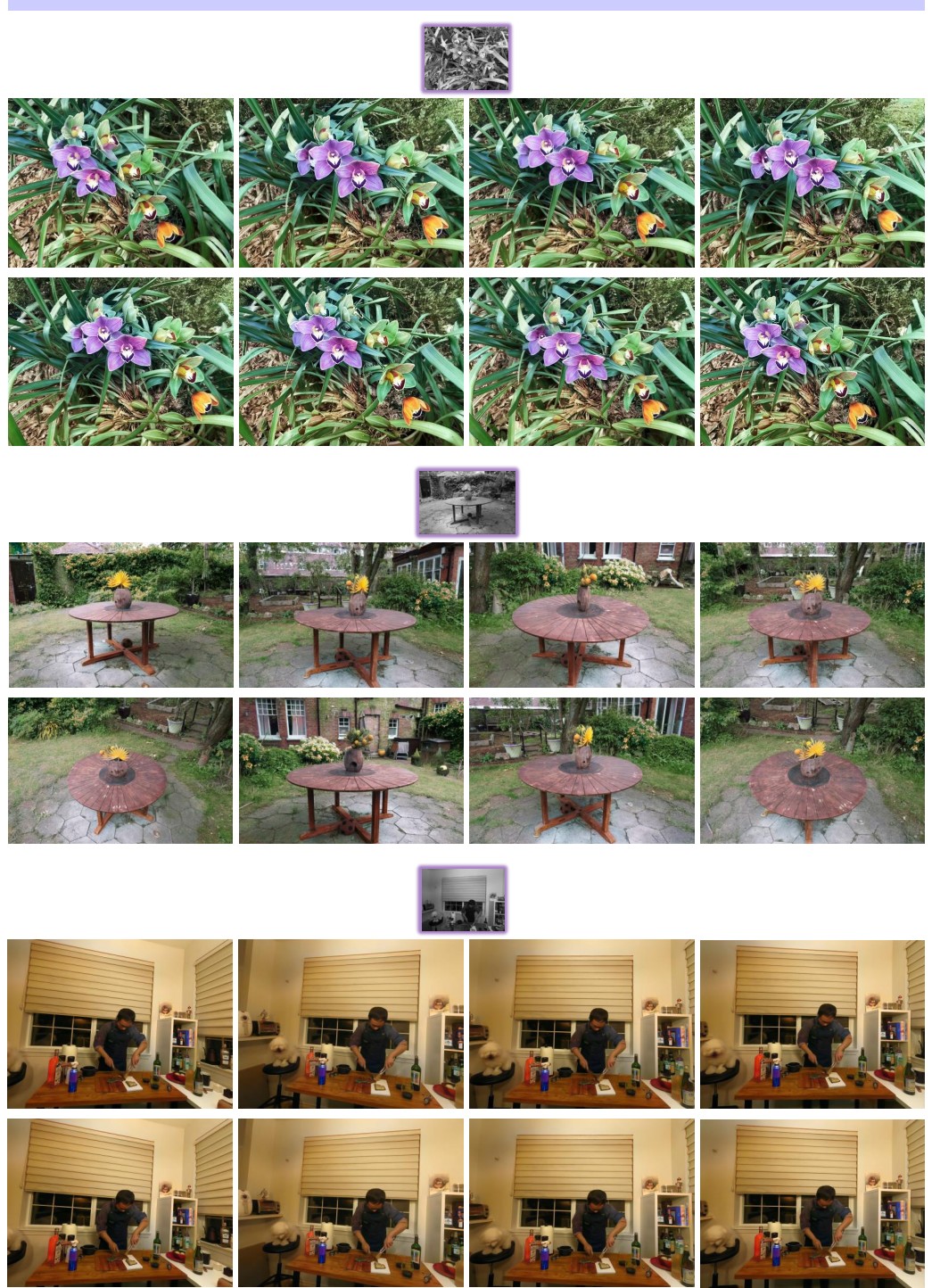

Figure 23: Visual results of automatic 3D scene colorization. From top to bottom are samples from the LLFF, Mip-NeRF 360, and DyNeRF datasets.

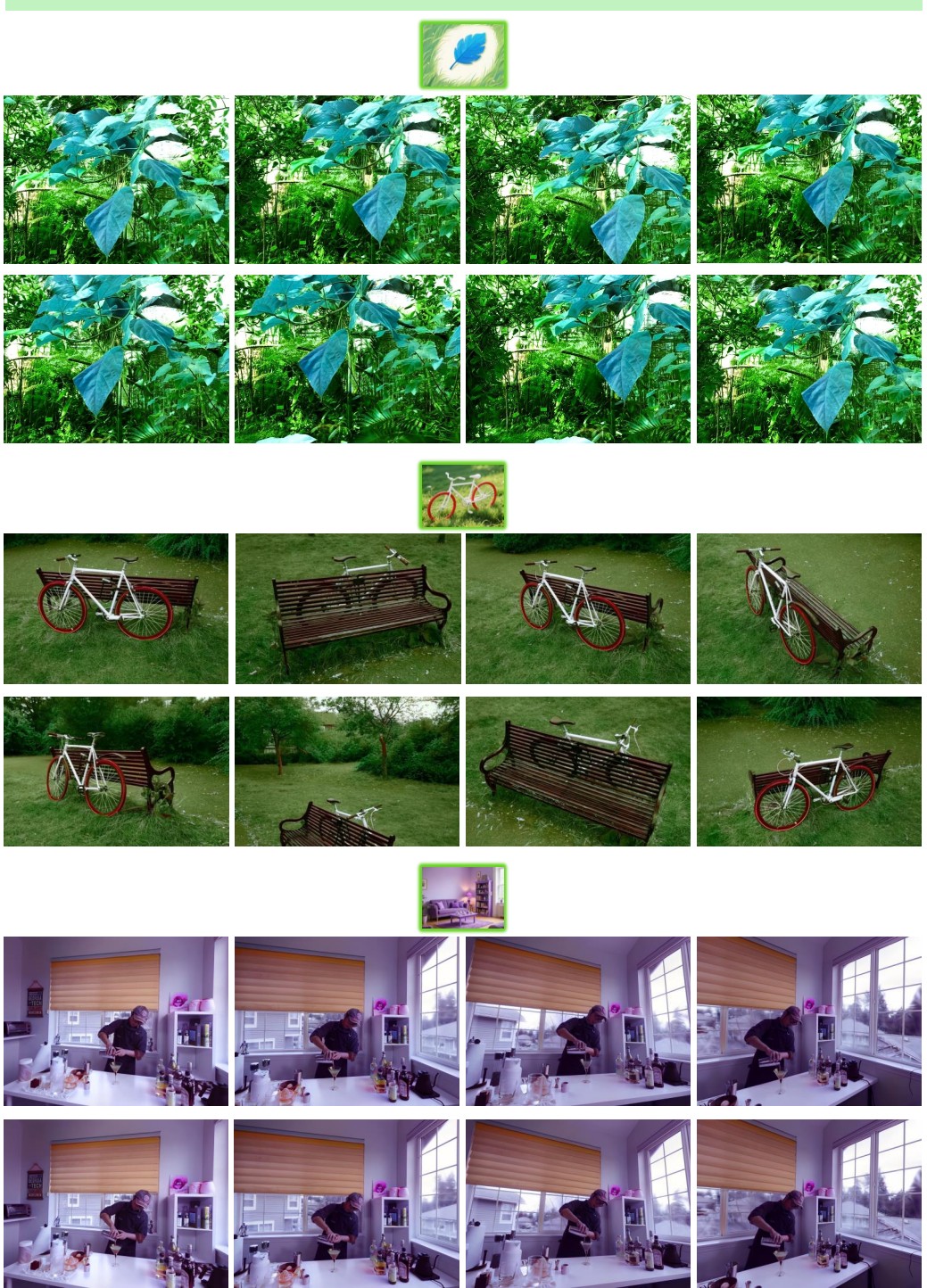

Figure 24: Visual results of reference-based 3D scene colorization. From top to bottom are samples from the LLFF, Mip-NeRF 360, and DyNeRF datasets.

