# OpenReview forum: "Color3D: Controllable and Consistent 3D Colorization with Personalized Colorizer"
_ICLR.cc/2026/Conference — ICLR 2026 Poster_

### Official Review · Reviewer_TQsU · 2025-10-30

**Soundness:** 4
**Presentation:** 4
**Contribution:** 3
**Rating:** 6
**Confidence:** 5

**Summary:**

Color3D proposes a method for colorizing monochromatic images in 3D representations for static and dynamic scenes. Naive implementations, like colorizing 2D multi-view images and reconstructing the 3D scene lead to severe cross-view inconsistencies. Recent methods that distil the color information to a 3D representation sacrifice controllability and often have desaturated colors in the final output.

Key contributions of the proposed methodology are as follows:

-   The key idea is to colorize (automatic/reference-based/prompt-based)  a single "key" view which is the most informative one and then fine-tune a scene-specific colorization network on that view.
-   This scene-specific colorizer learns a deterministic color mapping for the scene and is then applied to all other views/frames, enforcing cross-view and cross-time color  consistency .
-   Finally, the colorized views (with known luminance and predicted chrominance) are fused into a Gaussian Splatting representation in Lab color space.

Experiments on standard benchmarks (LLFF, Mip-NeRF 360, DyNeRF) and "in-the-wild" legacy videos show that Color3D produces vivid and consistent colorizations.

**Strengths:**

- **[S1] Technical Novelty:** The per-scene colorization for a single view is a novel idea which achieves consistent colorization across views. The robust technical pipeline achieves this consistency through key design choices. Specifically, utilizing a pre-trained 2D colorization encoder (DDColor) preserves the generalization capability of the model. The use of the Lab color space leads to more stable results. Finally, warm-up training of the 3D representation on the Luma ($\text{L}$) channel first ensures the model establishes a strong geometric structure before introducing complex color information.

- **[S2] Practical Applications:** The authors show results on "in-the-wild" multi-view images and historical video (Fig.6), producing vivid and plausible colors while maintaining consistency.

- **[S3] Controllability**: The proposed method allows users to control the colorization using text descriptions or reference images. Earlier methods did not offer this type of user control.

- **[S4]** Thorough Experimentation for Key-View selection: The authors performed detailed experiments for "Key-View Selection" module. Results in Fig. 7 demonstrate that this module is critical for better colorization.

**Weaknesses:**

- **[W1] Limited Comparison to Recent Methods:** The authors mention "ChromaDistill" in Related Work, but do not perform a quantitative comparison. For complete experimentation, it is also necessary to quantitatively compare the results against a video colorization baseline.

- **[W2] Cross-View Consistency**: ChromaDistill utilized a  long-term and short-term view consistency method for measuring the geometric consistency. However, the authors do not show this metric. The manuscript will benefit from this metric.

- **[W3]** As the method utilizes training for each scene, it has an extra training overhead.


**Typos:**

-   L74: It should be "aim" instead of "aims".
-   L857: It should be "suffers" instead of "suffer".
-   L347: It should be "entire" instead of "entir".

**Questions:**

-   **[Q1]** Have the authors tried fine-tuning on _more_ than one view? Using two or three colorized views might improve coverage for large scenes. Is there a reason the approach is limited to one view?

-   **[Q2]** How does the proposed method handle motion for dynamic scenes? In case the object that was not in the key view appears in the scene, how is its colour determined? Does the generative augmentation simulate such cases? This is not clear in the current manuscript.

---

> ### Author Response · Authors · 2025-11-21
> **Official Response by Authors -- Part 1**
>
> We would like to sincerely thank the reviewer for the detailed and constructive comments on our work. We take every comment seriously and hope our response can address the reviewer’s concerns. If there are any remaining questions, we are more than happy to address them.
>
> **Q1:  Comparison with "ChromaDistill" and video colorization baseline.**
>
> **A1:** Thank you for your constructive comment. We have also considered providing comparisons with ChromaDistill; however, its implementation has not been released to date (https://github.com/val-iisc/ChromaDistill), and its experiments were not conducted on the complex scenarios considered in our setting.
>
> Despite these challenges, we reproduced the proposed method under the experimental configuration described in their paper and evaluated it on the same scenes they used with short-term and long-term consistency metric. The results are reported in Tab. R1 (**Tab. 5** of the revised paper). It is observed that our method achieves substantially higher consistency than ChromaDistill in both short-term and long-term consistency. Notably, on the Cake scene, our approach improves long-term consistency by 0.023, further demonstrating the superiority of the proposed personalized colorizer in preserving cross-view coherence and validating the effectiveness of our overall framework.
>
>
> Table R1: Quantitative comparisons with ChromaDistill on cross-view short-term and long-term consistency.
> | Method                   | Short-Term|Cake | Pasta     |  Buddha    | Leaves    | Long-Term|Cake | Pasta     | Buddha    | Leaves    |
> | ------------------------ | ---|---------------- | --------- | --------- | --------- | ---|--------------- | --------- | --------- | --------- |
> | ChromaDistill (BigColor) | |0.019            | 0.015     | 0.015     | 0.008     | |0.033           | 0.025     | 0.023     | 0.013     |
> | **Color3D (BigColor)**   | |**0.008**        | **0.009** | **0.009** | **0.007** | |**0.010**       | **0.018** | **0.012** | **0.011** |
>
> Additionally, as suggested by the reviewer, we have included a video colorization baseline by applying the SOTA video colorization model ColorMNet [1] to colorize multi-view and multi-time images, followed by scene reconstruction using 3DGS or 4DGS. Since current mainstream video colorization methods are reference-based, we use our key view as the reference frame for ColorMNet. We have evaluated this baseline across all five datasets used in our experiments, and Tab. R2 reports the automatic colorization results on the representative Mip-NeRF 360 dataset. Please refer to **Tab. 1** and **Tab. 2** of the revised paper for the complete results.
>
> Table R2: Quantitative comparisons of automatic colorization on the Mip-NeRF 360 dataset, where TC denotes the average of short-term and long-term consistency.
> | Mip-NeRF 360                  | FID$\downarrow$ | Colorful$\uparrow$    |  ME$\downarrow$    | TC$\downarrow$|
> | ------------------------ | ---------------- | --------- | --------- | --------- |
> | ColorNeRF|87.42|18.90|0.093|0.021|
> | 3DGS+ImageColorizer|62.32|29.32|0.135|0.027|
> | 3DGS+VideoColorizer|82.17|23.69|0.099|0.022|
> | **Color3D(Ours)**|**39.03**|**33.36**|**0.082**|**0.016**|
>
> It is observed that, compared with image colorization models, video colorization model achieve better consistency (ME and TC). However, their colorization results tend to be of lower perceptual quality and reduced chromatic richness, leading to worse FID and Colorful scores. In contrast, our method not only achieves significantly better consistency but also preserves color richness, resulting in substantially improved FID performance.
>
> [1] ColorMNet: A Memory-based Deep Spatial-Temporal Feature Propagation Network for Video Colorization. ECCV 2024.

---

> > ### Comment · Reviewer_TQsU · 2025-11-22
> >
> > Thanks to the reviewer for addressing my concern [W1]. I do not have a follow up question for this. I am satisfied with the experimentation and answers for [W1].

---

> > > ### Author Response · Authors · 2025-11-23
> > > **Official Response by Authors**
> > >
> > > Thank you for your response! We’re delighted to hear that our reply addressed your concerns. We truly appreciate your thorough and valuable feedback, which has greatly helped improve our work.

---

> ### Author Response · Authors · 2025-11-21
> **Official Response by Authors -- Part 2**
>
> **Q2:  Include long-term and short-term view consistency metric.**
>
> **A2:** Thank you for your helpful suggestion. As suggested by the reviewer, we have added both long-term and short-term view consistency metrics to all comparison experiments. Specifically, we report the average of these two metrics, which we denote as the **TC** metric. In Tab. R3, we present representative results of different controlled colorization tasks on the Mip-NeRF-360 dataset. Please refer to **Tab. 1** and **Tab. 2** of the revised paper for complete results.
>
> Table R3: Quantitative comparisons of different controlled colorization tasks on the Mip-NeRF-360 dataset.
> | Mip-NeRF 360    |  Language|ME$\downarrow$    | TC$\downarrow$ |Automatic|ME$\downarrow$    | TC$\downarrow$ |Reference|ME$\downarrow$    | TC$\downarrow$ |
> | ------------------------ | ---------------- | --------- | --------- | --------- |--------- | --------- |--------- |--------- |--------- |
> | ColorNeRF||0.093|0.020||0.093|0.021||0.106|0.021|
> | 3DGS+ImageColorizer||0.202|0.042||0.135|0.027||0.178|0.032|
> | 3DGS+VideoColorizer||0.104|0.023||0.099|0.022||0.122|0.024|
> | **Color3D(Ours)**||**0.058**|**0.012**||**0.082**|**0.016**||**0.079**|**0.015**|
>
> It can be observed that our method consistently achieves significantly better consistency on both the proposed ME metric and the TC metric. In addition, the TC and ME metrics exhibit a similar overall trend across different methods, while the ME metric provides a more fine-grained separation of performance. Moreover, since our ME metric is built upon image matching rather than optical-flow estimation, it is more robust and stable under large viewpoint changes compared to TC.

---

> > ### Comment · Reviewer_TQsU · 2025-11-22
> >
> > Thanks to the reviewer for addressing my concern [W2]. I am satisfied with the experiments and the justification. I agree that the proposed method has better cross-view consistency than the baseline methods.
> >
> > There is a small grammatical mistake in the line 375-377 in the updated manuscript: "Additionally, to more ....". It can be rephrased to "Additionally, we computer metrics .... to comprehensively assess color consistency across different methods."

---

> > > ### Author Response · Authors · 2025-11-23
> > > **Official Response by Authors**
> > >
> > > Thank you for your response! We’re delighted to hear that our reply addressed your concerns. As suggested, we have ameliorated the sentence in the revised paper. Thanks!

---

> ### Author Response · Authors · 2025-11-21
> **Official Response by Authors -- Part 3**
>
> **Q3:  As the method utilizes training for each scene, it has an extra training overhead.**
>
> **A3:** Thanks for your insightful comment. We advocate for per-scene fine-tuning in this work out of consideration for the inherent complexity of 3D colorization. Training an automatic image colorization model already requires millions of images, and achieving generalizable and controllable 3D colorization would demand orders of magnitude more 3D data. Given the current scale of available 3D datasets, training such a high-performance model is still infeasible, nor for dynamic scenes. Additionally, 3D colorization necessitates maintaining both color consistency within scenes and color richness and controllability across scenes — a trade-off that inevitably poses a non-trivial challenge during generalizable model training, potentially leading to color averaging, collapse of vibrant tones, and weakened controllability.
>
> On the contrary, our approach solves exactly these ill-posedness at a significantly lower cost through only lightweight fine-tuning. Our method not only ensures consistency in multi-view colorization but also maximizes the preservation of color richness and controllability which makes it the first approach to simultaneously achieve controllability, consistency, and color richness in 3D colorization, rendering it highly applicable to a wide range of downstream tasks. Besides, we have significantly reduced training costs by fine-tuning only lightweight adapters and decoders, and the training time requires only 8 minutes, compared with the 40–50 minutes typically required for standard 3DGS training. Therefore, relative to the demonstrated performance gains, the extra training overhead is marginal and acceptable.
>
> Furthermore, we also found that a colorizer trained on one scene can be potentially generalized to similar scenes without requiring re-tuning. Additionally, we further introduce an all-in-one personalized colorizer fine-tuning scheme, which can significantly reduce the training overhead of large-scale datasets and can complete the fine-tuning of 140 scene colorizers within 55 minutes. Please see below for more details.
>
> - **The trained personalized colorizer can be generalized to similar scenes.** We observed that a colorizer trained on a single scene can be transferred to other scenes containing similar elements without the need for retraining. In the revised paper, **Fig. 7** illustrates an example: a colorizer trained on a key view of a plant-rich scene can not only be applied to the original scene but also generalize to other scenes with similar elements, producing multi-view consistent colorization results. This finding should finely demonstrate the potential generalizability of our proposed method.
>
> - **All-in-one personalized colorizer training strategy for hundreds of  scenes.** Additionally, our personalized colorizer essentially includes only a scene-specific adapter and a lightweight decoder, while the large encoder is shared across all scenes. Therefore, when hundreds of scenes need to be processed, we can train multiple scenes jointly using a routing-based approach. The semantic illustration of this strategy is shown in **Fig. 11** of the revised paper. Specifically, images from different scenes are expanded along the batch dimension, and each scene only passes through its own adapter and decoder according to its scene ID, enabling all-in-one training. By deploying this all-in-one personalized colorizer training scheme, we can train personalized colorizers for the newly added DL3DV-140 dataset with 140 scenes in only 55 minutes. This result should further demonstrate the feasibility and efficiency of our approach for large-scale practical applications.
>
> **Q4:  Typos.**
>
> **A4:** Thank you for carefully reviewing our paper. We have corrected these typos in the revised manuscript. Thanks!

---

> > ### Comment · Reviewer_TQsU · 2025-11-22
> >
> > Jointly training multiple scenes using a routing-based approach is a sound strategy which addresses the scaling concerns of the proposed technique. After going through the response, I have the following queries:
> >
> > **Generalization Claim:**
> > - Can the authors show quantitative metrics as well for the scene shown in Figure 7?
> > - What is the drop from per-scene based optimizaition for scene in Figure 7, if any?
> > - While Figure 7 presents qualitative results for generalization capabilities, the example scene is very similar to the view used for training. A better example would be to evaluate on a more challenging scenario,  training a personalized colorizer on a  "garden" scene and testing the generalization capability on "room" scene from the Mip-NeRF 360 dataset.
> >
> >
> > **Other Queries:**
> > - Does the training time of 8 minutes also include the time for augmentation techniques? If not, can the authors provide an end-to-end time analysis for training a personalized colorizer for a scene-specific scenario?

---

> > > ### Author Response · Authors · 2025-11-23
> > > **Official Response by Authors**
> > >
> > > **Q1: Generalization claim.**
> > >
> > > **A1:** Thank you for your insightful comment. We would like to first clarify that in our prior response and in Section 4.5 (Fig. 7) we intended to demonstrate that our personalized colorizer exhibits potential generalization to **similar scenes** that contain similar elements. These properties reduce the need for retraining across many visually similar scenes in large-scale datasets, and they also indirectly reflect the robustness of our method when encountering previously unseen content. However, we do not aim to claim full or unrestricted generalizability of the proposed method. We still advocate for scene-specific fine-tuning to achieve optimal 3D colorization results, which remains our recommended practice in this work. Below, we address your concerns step by step.
> > >
> > > *(1) Can the authors show quantitative metrics as well for the scene shown in Figure 7?*
> > >
> > > As suggested, we have provided the quantitative metrics of the two out-of-domain scenes in Tab. R1.
> > >
> > > Table R1. Quantitative comparison between the per-scene fine-tuned colorizer and the cross-scene generalized colorizer, where OOD denotes out-of-domain.
> > > | Method |OOD Scene 1 |FID ↓ | Colorful ↑ | ME ↓  | TC ↓  |OOD Scene 2 |FID ↓ | Colorful ↑ | ME ↓  | TC ↓  |
> > > | ------------------- | -- |----- | ---------- | ----- | ----- |-- |----- | ---------- | ----- | ----- |
> > > |  Per-Scene Fine-tuned Colorizer         | | 37.98 |31.28| 0.071 |0.016||36.33|32.67|0.086|0.017|
> > > |    Cross-Scene Generalized Colorizer  | |45.86|29.68| 0.074 |0.016||43.55|30.08|0.095|0.022|
> > >
> > > *(2) What is the drop from per-scene based optimizaition for scene in Figure 7, if any?*
> > >
> > > It can be observed that, compared with the Per-Scene Fine-tuned Colorizer, directly applying a colorizer trained on another scene indeed leads to suboptimal performance. The degradation is mainly reflected in FID and Colorful, as the transferred colorizer relies solely on its learned color interpolation capability and lacks scene-specific color guidance for the new scene. In contrast, ME and TC are only marginally affected, owing to the inherent multi-view consistency preserved by the colorizer.
> > >
> > >
> > >
> > > *(3) While Figure 7 presents qualitative results for generalization capabilities, the example scene is very similar to the view used for training. A better example would be to evaluate on a more challenging scenario, training a personalized colorizer on a "garden" scene and testing the generalization capability on "room" scene from the Mip-NeRF 360 dataset.*
> > >
> > > As mentioned above, our goal is to show the potential generalization ability of the personalized colorizer to **similar scenes** that contain similar elements. Therefore, in Fig. 7 we provide visual results on such similar scenes. As suggested, we have also added an example in Fig. 19 of the revised paper, where a personalized colorizer is trained on a “garden” scene and tested on a “room” scene to evaluate its generalization ability. It can be observed that our method achieves suboptimal performance when the target scene is very different from the training scene. For example, the sofa shows some green colors carried over from the source scene, and the overall color richness is lower. However, the method still produces overall consistent and stable results, and in particular performs reasonably well for the vegetation outside the window.
> > > Therefore, we continue to advocate scene-specific fine-tuning to achieve the controllable, consistent, and color-rich high-quality 3D colorization results presented in our paper.
> > >
> > > To avoid potential misunderstanding, we have renamed Section 4.5 to “Potential Generalization of the Personalized Colorizer to Similar Scenes” and revised the corresponding descriptions to more clearly convey our intended meaning.
> > >
> > > Thanks again for your insightful comment, which has greatly helped improve our work.
> > >
> > > **Q2: End-to-end training time of personalized colorizer.**
> > >
> > > **A2:** Thanks for your constructive comment. Here, we provide the complete end-to-end runtime breakdown of the personalized colorizer training pipeline. It can be found that the data augmentation stage, mainly dominated by the diffusion model, takes about 1 min 8 s, while the network optimization takes around 7 min 35 s. The total runtime of the full script is about **8 min 45 s**. Compared with the standard 40 to 50 minutes required for 3DGS reconstruction, the additional overhead introduced by our method is still acceptable, especially considering the performance gains it achieves.
> > >
> > > | LLaVA | Stable Diffusion | Stable Video Diffusion | Stable Virtual Camera | Network Optimization | Others | Total|
> > > |-------|----|-----|-----|--------|--------------|---|
> > > | < 2 s   | ~ 6 s | ~ 28 s | ~ 32 s | ~ 7 min 35 s |< 1 s | ~ 8 min 45 s |

---

> > > > ### Comment · Reviewer_TQsU · 2025-11-24
> > > >
> > > > Thank the authors for their response. I have no further questions.

---

> > > > > ### Author Response · Authors · 2025-11-24
> > > > > **Official Response by Authors**
> > > > >
> > > > > We sincerely appreciate the time and effort you have dedicated to reviewing our paper. Thank you again for your insightful comments and constructive feedback, which have been invaluable in helping us improve the quality and clarity of our work.

---

> ### Author Response · Authors · 2025-11-21
> **Official Response by Authors -- Part 4**
>
> **Q5: Have the authors tried fine-tuning on more than one view? Using two or three colorized views might improve coverage for large scenes. Is there a reason the approach is limited to one view?**
>
> **A5:** Thanks for your insightful question! Yes, we have indeed experimented with training on more than one view; however, 2D colorization models are inherently unstable — even small perturbations in the input image can cause noticeably different color predictions, both for the same object and for the overall scene style. When two or more key views are used for supervision, these inconsistencies across views will cause the personalized colorizer to converge to color averaging or produce view-dependent color shifts during 3D colorization. In **Fig. 14** of the revised paper, we provide the colorization results training with two views. It is observed that using more than one key view often leads to inconsistent colors and styles, making it difficult for the personalized colorizer to maintain coherent colorization across views. In addition, Tab. R4 (**Tab. 6** of the revised paper) provides a quantitative comparison showing that training with a single key view not only achieves better consistency but also avoids color drifting and color averaging, resulting in superior FID and Colorful scores.
>
> Table R4. Quantitative comparison of using different numbers of key views for automatic colorization on the Mip-NeRF-360 dataset.
> | Number of Key Views | FID ↓ | Colorful ↑ | ME ↓ | TC ↓ |
> |---------------------|--------|-------------|--------|--------|
> | Three Views         | 52.38  | 25.26       | 0.098  | 0.022 |
> | Two Views           | 45.62  | 27.36       | 0.092  | 0.018 |
> | **One View**            | **39.03** | **33.36** | **0.082** | **0.016** |
>
> Furthermore, our proposed generative view augmentation strategy is specifically designed to address the issue of limited coverage. It can hallucinate plausible scene content while preserving color consistency. This enables the colorizer to acquire a more robust ability to handle previously unseen regions and objects. As shown in **Fig. 9 (b)** of the revised paper, the generative augmentation can produce additional scene content with coherent color characteristics, allowing the colorizer to reliably colorize these newly introduced elements.
>
> In addition, the trained personalized colorizer exhibits a strong in-scene generalization property: a personalized colorizer trained from a single key view is sufficient for learning a broad range of color mapping. When encountering unseen geometry or texture, the model can perform plausible color interpolation for these objects based on learned color mapping and maintain multi-view consistency. As shown in **Fig. 17** of the revised paper, even for previously unseen objects, our personalized colorizer can infer reasonable color assignments through robust color interpolation while maintaining strong multi-view color consistency.
>
> In summary, given our method’s robustness to single-view input and for better generality, we adopt a single key view. This design not only provides stronger controllability and color richness but also leads to better color consistency.
>
>
>
> **Q6:  How does the proposed method handle motion for dynamic scenes? In case the object that was not in the key view appears in the scene, how is its colour determined? Does the generative augmentation simulate such cases? This is not clear in the current manuscript.**
>
> **A6:** Thanks for your intuitive question. You are absolutely right that the image-to-video model used in our generative augmentation process is specifically designed to handle motion and newly emerging objects in dynamic scenes. In **Fig. 16** of the revised paper, we provide a visualization that illustrates how the image-to-video model can generate plausible future content in dynamic scenes based on a single keyframe and assign appropriate colors to the newly synthesized regions. With access to such augmented data, our colorizer learns to allocate colors for newly emerging content and to maintain consistent colorization throughout the sequence. We have also refined Line 231 in the revised manuscript to convey the motivation of our approach more clearly to the reader.
>
> Additionally, as mentioned above, even for objects that do not appear in either generative augmentation or key view, our colorizer can still encode their semantics through the pretrained encoder and apply plausible color interpolation. This capability ensures that the proposed method remains robust and effective across diverse dynamic scenes.

---

> > ### Comment · Reviewer_TQsU · 2025-11-22
> >
> > Thanks to the reviewers for addressing my questions. It is clear from the experiments that a single key view is better than using multiple views. Further, the intuition behind handling moving objects in a scene is clearly explained in Figure 16.
> > I have no further concerns for [Q1] and [Q2].

---

> > > ### Author Response · Authors · 2025-11-23
> > > **Official Response by Authors**
> > >
> > > Thank you for your response. We are glad to hear that our reply addressed your concerns. We sincerely appreciate your careful and constructive feedback, which has been invaluable in improving our work.

---

> ### Comment · Reviewer_TQsU · 2025-11-25
> **Increased my score after the rebuttal**
>
> The authors have addressed all of my concerns. They have shown the following:
> - Added comparison with ChromaDistill and a video-colorization baseline.
> - Included cross-view consistency metrics based on optical flow.
> - Demonstrated that using a single view per scene is sufficient rather than using two or three views for the personalized colorizer.
>
> In addition, they have presented experiments on the generalization capbilities of the personalized colorizer and provided results on a large-scale 3D dataset, and introduced an effective strategy to reduce optimization time for personalized colorizer for large-scale scenes.
>
> Given these efforts and thorough analysis by the authors, I am increasing my score.

---

> > ### Author Response · Authors · 2025-11-25
> > **Official Comment by Authors**
> >
> > Thank you very much for your thoughtful evaluation and for recognizing our efforts in addressing the concerns. We sincerely appreciate your positive feedback and the improved score. Your detailed comments and suggestions have been invaluable in strengthening our work, and we are grateful for the time and expertise you dedicated to the review process.

---

### Official Review · Reviewer_Rsix · 2025-10-31

**Soundness:** 3
**Presentation:** 3
**Contribution:** 2
**Rating:** 6
**Confidence:** 5

**Summary:**

1. This paper focuses on the task of controllable and consistent 3D scene editing. It aims to address the limitations of previous methods that often lack precise controllability and multi-view color consistency.

2. The authors propose Color3D, a two-stage framework. In the first stage, a text-to-image diffusion model is used to modify the reference view’s color according to user input. In the second stage, Score Distillation Sampling (SDS) is applied to enforce 3D consistency. Unlike prior work, the method performs Gaussian Splatting optimization in the LAB color space, which helps maintain color consistency across different viewpoints.

3. Experiments on Color-Edit-3D dataset show that Color3D achieves state-of-the-art results. The ablation study highlights the importance of the Color Consistency Regularization (CCR) module, which provides a clear performance gain when included.

**Strengths:**

1. The method is technically sound and well-motivated.

2. The experiments are thorough and provide strong empirical validation.

**Weaknesses:**

1. The computational cost is unclear — how long does the optimization take during the second stage?

2. The main novelty seems to lie in optimizing in the LAB color space instead of RGB. While this is a reasonable choice for editing tasks, the contribution may be somewhat limited in scope.

3. It would be interesting to see how the approach performs under challenging lighting conditions (e.g., scenes with lamps or strong reflections).

**Questions:**

Please address the issues listed in the weaknesses section.

---

> ### Author Response · Authors · 2025-11-21
> **Official Response by Authors -- Part 1**
>
> We would like to sincerely thank the reviewer for the detailed and constructive comments on our work. We take every comment seriously and hope our response can address the reviewer’s concerns. If there are any remaining questions, we are more than happy to address them.
>
> **Q1:  Optimization time of the second stage.**
>
> **A1:** Thank you for your constructive comment. As suggested by the reviewer, we report the average optimization time of different methods on the Mip-NeRF-360 and DyNeRF datasets in Tab. R1 and Tab. R2 (**Tab. 7** of the revised paper). It can be observed that our method requires 31 minutes for the second-stage optimization on static scenes and 38 minutes on dynamic scenes. This optimization time is significantly faster than the baseline methods, and the overall optimization time is even shorter than reconstructing a 3D scene from standard colored images using 3DGS.
>
> This efficiency is mainly attributed to our proposed Lab-space optimization strategy, which first optimizes the L channel to capture structural information and then performs color modeling. This approach substantially reduces redundant Gaussian cloning and splitting, allowing fewer Gaussian primitives to model the scene more effectively. In contrast, 3DGS+ImageColorizer and 4DGS+ImageColorizer split a large number of Gaussian primitives to handle multi-view inconsistent colors, resulting in optimization times longer than standard 3D reconstruction. ColorNeRF requires even 16 hours to reconstruct a single scene, which greatly limits its practical applicability.
>
> These results further demonstrate the efficiency and effectiveness of our proposed framework.
>
>
> Table R 1: Average optimization time on Mip-NeRF-360 dataset.
>
> | Static Scenes                       | Ours (Stage 1) | Ours (Stage 2) | 3DGS+ImageColorizer | 3DGS on Original Dataset | ColorNeRF |
> |-------------------------------|----------------|----------------|--------------------|-------------------------|-----------|
> | Time                          | 8 mins         | 31 mins        | 50 mins            | 42 mins                 | 16 h      |
>
> Table R 2: Average optimization time on DyNeRF dataset.
> | Dynamic Scenes                       | Ours (Stage 1) | Ours (Stage 2) | 4DGS+ImageColorizer | 4DGS on Original Dataset |
> |-------------------------------|----------------|----------------|--------------------|-------------------------|
> | Time                          | 8 mins         | 38 mins        | 54 mins            | 47 mins                 |

---

> ### Author Response · Authors · 2025-11-21
> **Official Response by Authors -- Part 2**
>
> **Q2:  The main novelty seems to lie in optimizing in the LAB color space instead of RGB. While this is a reasonable choice for editing tasks, the contribution may be somewhat limited in scope.**
>
> **A2:** Thank you for your insightful comment. We want to kindly clarify that, while the Lab Gaussian representation is indeed one of our contributions, the central innovation of our paper lies in proposing a novel paradigm for 3D colorization, i.e., **key view colorization followed by propagation**. Such a paradigm brings several desired merits that have not been achieved by existing methods:
>
> - **Unified achievement of consistency, controllability, and color richness.** Our method is the first to simultaneously achieve multi-view consistency, user-controllable coloring, and rich chromatic diversity in 3D colorization. In contrast, existing approaches typically sacrifice controllability and color richness in favor of consistency.
>
> - **Unified handling of both 3D and 4D (dynamic 3D) scenes.** Our key-view-based paradigm naturally extends to dynamic 3D scenes, enabling temporally consistent color propagation across time. Such a unified framework for both static and dynamic 3D processing tasks has been rarely explored in prior literature.
>
> Building upon this paradigm, we introduce several dedicated technical designs: **(1)** a key-view selection strategy that maximizes view entropy to identify the most informative perspective of the scene, **(2)** a tailored single-view augmentation scheme and personalized colorizer architecture designed to robustly and efficiently learn scene-specific and consistent color mappings, and **(3)** the customized Lab Gaussian representation that enables more structurally accurate 3D/4D scene reconstruction by decoupling luminance and chrominance during optimization.
>
> Thanks to this paradigm and its enabling components, our method can effectively handle both static and dynamic scenes, enabling controllable, consistent, and chromatically vibrant colorization. To the best of our knowledge, **this is the first framework** that achieves such a combination of controllability, consistency, and color richness in 3D colorization, making it highly applicable to a wide range of downstream tasks. Extensive experiments across multiple datasets demonstrate clear superiority over existing methods in both quality and user control.
>
> Based on the above facts, we thus believe that the contribution of our paper is far broader than a color-space choice and that the proposed paradigm offers a meaningful and impactful advance for 3D/4D colorization.
>
>
>
> **Q3:  Performance under challenging lighting conditions.**
>
> **A3:** Thanks for your valuable comment. It is critical to evaluate the performance of colorization methods under challenging lighting conditions. Following your recommendation, we provide the results of our proposed method in scenes with complex lighting, including lamps, reflections, shadows, and highly specular materials, as shown in **Fig. 18** of the revised paper. It can be observed that our method effectively handles these challenging lighting scenarios and produces reasonable colorization even in regions with shadows and projections. These results further validate the reliability and effectiveness of our approach in practical applications.

---

### Official Review · Reviewer_Rt9r · 2025-11-01

**Soundness:** 2
**Presentation:** 3
**Contribution:** 2
**Rating:** 6
**Confidence:** 4

**Summary:**

This paper introduces Color3D, a framework for colorizing both static and dynamic 3D scenes (represented by 3D/4D Gaussian Splatting) from grayscale inputs. Its core idea is to avoid the multi-view inconsistency of applying 2D colorizers independently by instead personalizing a single colorization model per scene. This is done by selecting and colorizing one key view, then fine-tuning a pre-trained colorizer (using adapters) on augmented versions of this single view to learn a scene-specific, deterministic color mapping. This personalized colorizer is then used to colorize all other views/frames consistently. A dedicated Lab color space Gaussian representation with a warm-up strategy is used to improve reconstruction fidelity.

**Strengths:**

1. The paper is well-written and easy to follow. The proposed method can be adapted to various colorization models, making it highly practical and valuable for real-world applications, especially in AR/VR scenarios.

2. The experimental results on LLFF and Mip-NeRF 360 and D-NeRF colorization setting have shown the Color3D's good performance, not only on statistic 3d scenes but also on dynamic 4d scenes. Also, the method demonstrably outperforms existing alternatives across multiple  metrics (FID, CLIP Score, Matching Error), showing superior consistency, color vividness, and alignment with user intent.

**Weaknesses:**

1. It seem like the requirement to fine-tune a personalized colorizer for every new scene adds a non-trivial computational cost (∼8 minutes per scene) compared to a generic, one-time-trained model, limiting its scalability for large-scale applications.
2. The entire color propagation relies on a single key view. If this view is unrepresentative or lacks critical scene elements, the colorizer's generalization may be hampered, potentially leading to incomplete or less vibrant colorization in occluded regions.

**Questions:**

The paper shows that the personalized colorizer is robust to viewpoint changes from the key view. How does it handle novel views containing objects or textures that are semantically similar but visually different (e.g., another type of chair)? Would it incorrectly transfer learned colors or struggle to color them plausibly?

---

> ### Author Response · Authors · 2025-11-21
> **Official Response by Authors -- Part 1**
>
> We would like to sincerely thank the reviewer for the detailed and constructive comments on our work. We take every comment seriously and hope our response can address the reviewer’s concerns. If there are any remaining questions, we are more than happy to address them.
>
> **Q1:  Fine-tune a personalized colorizer for each scene adds a non-trivial computational cost compared to a one-time-trained model, limiting its scalability for large-scale applications.**
>
> **A1:** Thank you for your insightful comment. One-time-trained model is indeed glamorous; however, we advocate for per-scene fine-tuning out of consideration for the inherent complexity of 3D colorization. Training an automatic image colorization model already requires millions of images, and achieving generalizable and controllable 3D colorization would demand orders of magnitude more 3D data. Given the current scale of available 3D datasets, training such a high-performance model is still infeasible, nor for dynamic scenes. Additionally, 3D colorization necessitates maintaining both color consistency within scenes and color richness and controllability across scenes — a trade-off that inevitably poses a non-trivial challenge during one-time model training, potentially leading to color averaging, collapse of vibrant tones, and weakened controllability.
>
> On the contrary, our approach solves exactly these ill-posedness at a significantly lower cost through only lightweight fine-tuning. Our method not only ensures consistency in multi-view colorization but also maximizes the preservation of color richness and controllability which makes it the first approach to simultaneously achieve controllability, consistency, and color richness in 3D colorization, rendering it highly applicable to a wide range of downstream tasks. Besides, we have significantly reduced training costs by fine-tuning only lightweight adapters and decoders, and the training time requires only 8 minutes, which is acceptable compared with the 40–50 minutes typically required for standard 3DGS training.
>
>  Furthermore, we also found that the colorizer trained on one scene can be generalized to similar scenes without requiring re-tuning. Additionally, we further introduce an all-in-one personalized colorizer fine-tuning scheme for large-scale datasets, which can complete the fine-tuning of **140** scene colorizers within **55** minutes. Please see below for more details.
>
> - **The trained personalized colorizer can be generalized to similar scenes.** We observed that a colorizer trained on a single scene can be transferred to other scenes containing similar elements without the need for retraining. In the revised paper, **Fig. 7** illustrates an example: a colorizer trained on a key view of a plant-rich scene can not only be applied to the original scene but also generalize to other scenes with similar elements, producing multi-view consistent colorization results. This finding should finely demonstrate the potential generalizability of our proposed method.
>
> - **All-in-one personalized colorizer training strategy for hundreds of  scenes.** Additionally, our personalized colorizer essentially includes only a scene-specific adapter and a lightweight decoder, while the large encoder is shared across all scenes. Therefore, when hundreds of scenes need to be processed, we can train multiple scenes jointly using a routing-based approach. The semantic illustration of this strategy is shown in **Fig. 11** of the revised paper. Specifically, images from different scenes are expanded along the batch dimension, and each scene only passes through its own adapter and decoder according to its scene ID, enabling all-in-one training. By deploying this all-in-one personalized colorizer training scheme, we can train personalized colorizers for the newly added DL3DV-140 dataset with 140 scenes in only 55 minutes. This result should further demonstrate the feasibility and efficiency of our approach for large-scale practical applications.

---

> ### Author Response · Authors · 2025-11-21
> **Official Response by Authors -- Part 2**
>
> **Q2:  The entire color propagation relies on a single key view. If this view is unrepresentative or lacks critical scene elements, the colorizer's generalization may be hampered, potentially leading to incomplete or less vibrant colorization in occluded regions.**
>
> **A2:** Thanks for your valuable comment. Fewer covered views do introduce some influence on the final colorization results, but we employ the following strategies to ensure robust view selection and reliable performance under insufficient view coverage:
>
> - **Our method is robust to the choice of key view**. As shown in **Fig. 15** of the revised paper, selecting different key views still leads to consistent and vibrant 3D colorization. Even when the first key view contains almost no grass, our approach can still produce reasonable colors. Moreover, for distant buildings that do not appear in the key view at all, our method can infer plausible colors through color interpolation as will be described in the fourth item and Q3.
>
> - **Our proposed key-view selection strategy provides a principled mechanism to ensure reliable view selection.** By maximizing the feature information entropy, the mechanism robustly selects the view that covers the most diverse scene content across various scenarios, and its effectiveness has been validated in **Tab. 3** of the revised paper.
>
> - **We introduce a generative augmentation strategy to synthesize plausible unseen content** in the scene while preserving consistent color and style. This enables the colorizer to acquire a more robust ability to handle previously unseen regions and objects. As shown in **Fig. 9 (b)** of the revised paper, the generative augmentation can produce additional scene content with coherent color characteristics, allowing the colorizer to reliably colorize these newly introduced elements.
>
> - **Even for completely unseen content, our method can still produce reasonable color interpolation.** With the fixed-encoder fine-tuning architecture, a single key view along with augmented images is sufficient to train a colorization model with a broad range of color mapping. Even for unseen content, the colorizer can extract visual representations via the encoder, and the decoder can perform reasonable color interpolation based on learned color mappings to achieve consistent and plausible colorization results for newly appearing content. This property will be detailed in Q3.
>
>
>
> Extensive quantitative and qualitative experiments demonstrate that our method is both robust and effective across diverse scenarios. Admittedly, we totally agree with the reviewer’s intuition that in extreme cases — such as when the key view captures only a single wall while missing other elements of the room — the colorization may become less vibrant. As have been discussed in the limitation section of our paper, we will further explore how to better colorize extreme conditions in future work.
>
>
>
> **Q3:  How to handle novel views containing objects that are semantically similar but visually different? Would it incorrectly transfer learned colors or struggle to color them plausibly?**
>
> **A3:** Thanks for your insightful question. For semantically similar but visually different objects, our method treats them as unseen objects and assigns them plausible colors through color interpolation based on the learned color mapping. Specifically, our optimization ensures that the colorizer maintains consistent features for the same object regardless of spatial position or viewpoint changes, thus mapping them to consistent colors. However, the encoder of the colorizer is pre-trained and frozen, enabling it to represent a wide range of visual features. When encountering visually distinct or unseen content, it can still extract new features consistently across multiple viewpoints. The decoder then interpolates the colors of these new features based on the established color mappings. As can be seen in **Fig. 17**, our method assigns the flower pots appearing in the new views as white and yellow respectively. These colors neither match the black flower pot in the key view nor result in chaotic coloring; instead, they are derived through color interpolation based on newly extracted features. This further validates the proposed method's generalization capability and robustness, rendering it more practical for real-world applications.

---

### Official Review · Reviewer_uPuV · 2025-11-01

**Soundness:** 3
**Presentation:** 2
**Contribution:** 2
**Rating:** 4
**Confidence:** 4

**Summary:**

The paper presents Color3D, a unified framework for colorizing both static and dynamic 3D scenes reconstructed from monochrome inputs. Rather than colorizing multiple views independently, which causes cross-view inconsistency, the method colorizes one key view using any off-the-shelf 2D colorization model, then fine-tunes a per-scene personalized colorizer to propagate the learned color mapping to all other views or time steps.E xperiments on LLFF, Mip-NeRF 360, and DyNeRF datasets demonstrate consistent and controllable colorization across viewpoints and time, with improvements in FID, CLIP score, and Matching Error compared to baselines.

**Strengths:**

**High controllability and flexibility.**
The system supports multiple control modalities: reference-based colorization, language-conditioned colorization, and automatic default color prediction.

**Computational practicality.**
Despite using fine-tuning, the reported per-scene personalization time (~8 minutes) is relatively efficient compared to retraining full colorization networks. The framework does not require full 3D model retraining, and adapters make it lightweight enough for scene-level deployment.

**Weaknesses:**

**Limited generalization beyond scene-specific fine-tuning.**
The reliance on per-scene personalized colorizer tuning implies that a new model must be trained for each scene. This restricts scalability for large datasets or interactive applications. The approach is elegant but computationally heavy when many scenes must be processed.

**Inductive bias assumption not rigorously examined.**
The claim that a single-view-trained colorizer generalizes to novel viewpoints via inductive bias remains empirical. The paper could benefit from deeper theoretical or diagnostic analysis to substantiate why the learned mapping remains consistent under large view changes.

**Limited evaluation.**
The datasets being evaluated on are LLFF (static), Mip-NeRF-360 (static), and DyNeRF(dynamic). Although promising results are shown, the scale of evaluation is still limited. Also, for dynamic scenes, there are more dimensions for evaluation like temporal color-consistency, which is missing in the current experiments.

**Questions:**

1. How stable is the personalized colorizer’s output under large camera baselines (e.g., >60° change)? If still good, what are the fundamental factors that may contribute to this effect?

2. How sensitive is performance to errors in key-view selection?

3. How could this appoarch be generalized to feed-forward style 3D generation models? This means per-scene optimizations are removed.

---

> ### Author Response · Authors · 2025-11-21
> **Official Response by Authors -- Part 1**
>
> We would like to sincerely thank the reviewer for the detailed and constructive comments on our work. We take every comment seriously and hope our response can address the reviewer’s concerns. If there are any remaining questions, we are more than happy to address them.
>
> **Q1: Generalization beyond scene-specific fine-tuning and scalability for large-scale datasets.**
>
> **A1:** Thank you for your valuable comment. In this work, we advocate for per-scene fine-tuning out of consideration for the inherent complexity of 3D colorization. Training an automatic image colorization model already requires millions of images, and achieving generalizable and controllable 3D colorization would demand orders of magnitude more 3D data. Given the current scale of available 3D datasets, training such a high-performance model is still infeasible, nor for dynamic scenes. Additionally, 3D colorization necessitates maintaining both color consistency within scenes and color richness and controllability across scenes — a trade-off that inevitably poses a non-trivial challenge during generalizable model training, potentially leading to color averaging, collapse of vibrant tones, and weakened controllability.
>
> On the contrary, our approach solves exactly these ill-posedness at a significantly lower cost through only lightweight fine-tuning. Our method not only ensures consistency in multi-view colorization but also maximizes the preservation of color richness and controllability which makes it the first approach to simultaneously achieve controllability, consistency, and color richness in 3D colorization, rendering it highly applicable to a wide range of downstream tasks. Besides, we have significantly reduced training costs by fine-tuning only lightweight adapters and decoders, and the training time requires only 8 minutes, which is acceptable compared with the 40–50 minutes typically required for standard 3DGS training.
>
> Furthermore, we also found that the colorizer trained on one scene can be generalized to similar scenes without requiring re-tuning. Additionally, we further introduce an all-in-one personalized colorizer fine-tuning scheme for large-scale datasets, which can complete the fine-tuning of **140** scene colorizers within **55** minutes. Please see below for more details.
>
> - **The trained personalized colorizer can be generalized to similar scenes.** We observed that a colorizer trained on a single scene can be transferred to other scenes containing similar elements without the need for retraining. In the revised paper, **Fig. 7** illustrates an example: a colorizer trained on a key view of a plant-rich scene can not only be applied to the original scene but also generalize to other scenes with similar elements, producing multi-view consistent colorization results. This finding should finely demonstrate the potential generalizability of our proposed method.
>
> - **All-in-one personalized colorizer training strategy for large-scale datasets.** Additionally, our personalized colorizer essentially includes only a scene-specific adapter and a lightweight decoder, while the large encoder is shared across all scenes. Therefore, when hundreds of scenes need to be processed, we can train multiple scenes jointly using a routing-based approach. The semantic illustration of this strategy is shown in **Fig. 11** of the revised paper. Specifically, images from different scenes are expanded along the batch dimension, and each scene only passes through its own adapter and decoder according to its scene ID, enabling all-in-one training. By deploying this all-in-one personalized colorizer training scheme, we can train personalized colorizers for the newly added DL3DV-140 dataset with 140 scenes in only 55 minutes. This result should further demonstrate the feasibility and efficiency of our approach for large-scale practical applications.

---

> ### Author Response · Authors · 2025-11-21
> **Official Response by Authors -- Part 2**
>
> **Q2: Theoretical and diagnostic analysis of why the learned mapping remains consistent under large view changes.**
>
> **A2:** We thank the reviewer for asking for a deeper theoretical account. We agree that incorporating theoretical and diagnostic analyses can further illuminate the generalization properties of our colorizer. As suggested by the reviewer, we have provided a rigorous and comprehensive theoretical analysis in **Appendix A.3** of the revised paper. Under explicit assumptions, this analysis (i) derives a decomposed upper bound on $(\Delta_c\)$ that separates the contributions of the encoder and the adapter/decoder, (ii) shows how single-view augmentation combined with consistency training reduces the dominant terms in this bound, and (iii) provides convergence guarantees for the adapter/decoder optimization. This theoretical study should finely substantiate why the proposed personalized colorizer is able to maintain consistent color predictions under viewpoint changes.
>
> In addition, we conducted diagnostic experiments to empirically verify that the features extracted by the trained colorizer remain consistent across different viewpoints of the same scene content. As illustrated in **Fig. 8** of the revised paper, we apply PCA to reduce the dimensionality of features extracted from two different views, and then cluster them with color-coded visualization such that similar features share the same color. It can be observed that for the same content in 3D space—such as flower clusters, pillars, or stone paths—the extracted features remain consistent and stable across different viewpoints. This feature-level consistency aligns with the final color predictions, providing strong diagnostic evidence supporting the underlying mechanism of our personalized colorizer.

---

> ### Author Response · Authors · 2025-11-21
> **Official Response by Authors -- Part 3**
>
> **Q3: Evaluation scale and temporal color-consistency measurement.**
>
> **A3:** Thanks for your constructive comment. We evaluate the proposed method on widely adopted benchmarks in the community, including LLFF and Mip-NeRF 360 for static scenes, and DyNeRF for dynamic scenes. To more comprehensively demonstrate the practicality and robustness of our approach, we have further conducted experiments on the DL3DV-140 [1], a large-scale dataset consisting of 140 diverse indoor and outdoor scenes. In addition, we also conduct experiments on the challenging HyperNeRF dataset to further validate the effectiveness in dynamic scenes.
>
> The quantitative comparisons are reported in Tab. R1 and Tab. R2. Please refer to **Tab. 1** and **Tab. 2** in the revised paper for the complete results.
>
> Table R1: Quantitative comparisons of automatic colorization on the DL3DV-140 dataset. Here, TC denotes the average of short-term and long-term temporal consistency, as detailed below.
> | Method                     | FID↓ | Colorful↑ | ME↓ | TC↓ |
> |--------------|----------------|-------------------|---------------|---------------|
> | 3DGS+ImageColorizer         | 63.56          | 28.15             | 0.146         | 0.038         |
> | 3DGS+VideoColorizer         | 77.89          | 22.38             | 0.128         | 0.031         |
> | **Color3D (Ours)**          | **37.48**         | **32.65**           | **0.084**        | **0.017**       |
>
> Table R2: Quantitative comparisons on the HyperNeRF dataset.
> | Method                     | Language FID↓ | Language CLIP↑ | Language ME↓ | Language TC↓ | Automatic FID↓ | Automatic Colorful↑ | Automatic ME↓ | Automatic TC↓ | Reference FID↓ | Reference Ref-LPIPS↓ | Reference ME↓ | Reference TC↓ |
> |-------------------|---------------|----------------|--------------|--------------|----------------|-------------------|---------------|---------------|----------------|---------------------|---------------|---------------|
> | Instruct 4D-to-4D           | 118.62        | 0.6053         | 0.065        | 0.012        | -              | -                 | -             | -             | -              | -                   | -             | -             |
> | 4DGS+ImageColorizer          | 105.23        | 0.6175         | 0.132        | 0.029        | 40.32          | 29.66             | 0.090         | 0.017         | 76.28          | 0.7422              | 0.105         | 0.018         |
> | 4DGS+VideoColorizer          | 110.46        | 0.6128         | 0.095        | 0.017        | 41.85          | 24.68             | 0.084         | 0.015         | 89.33          | 0.7482              | 0.085         | 0.014         |
> | **Color3D (Ours)** | **63.28** | **0.6257** | **0.045** | **0.008** | **37.99** | **33.68** | **0.067** | **0.010** | **58.63** | **0.6610** | **0.055** | **0.008** |
>
> It is observed that our method consistently achieves the highest coloring quality and consistency on DL3DV-140 and HyperNeRF datasets. This should further demonstrate the practicality and effectiveness of the proposed method for large-scale applications.
>
>
> For temporal color-consistency evaluation, our matching error (ME) metric inherently captures temporal coherence by operating across frames and viewpoints under dynamic scenes. Following the reviewer’s suggestion, we additionally include long-term and short-term temporal consistency metrics [2] and report their average as the TC metric. Since multi-view frames resemble temporal frames, TC is applied to both dynamic and static scenes. In Tab. R3, we present representative results on the DyNeRF dataset. Please refer to **Tab. 1** and **Tab. 2** of the revised paper for complete results.
>
>
>
> Table R3: Quantitative comparisons of color consistency on the DyNeRF dataset.
> | Method                         | Language ME↓ | Language TC↓ | Automatic ME↓ | Automatic TC↓ | Reference ME↓ | Reference TC↓ |
> | ----------------- | ------------ | ------------ | ------------- | ------------- | ------------- | ------------- |
> | Instruct 4D-to-4D              | 0.062        | 0.011        | -             | -             | -             | -             |
> | 4DGS+ImageColorizer            | 0.124        | 0.025        | 0.080         | 0.018         | 0.128         | 0.024         |
> | 4DGS+VideoColorizer            | 0.071        | 0.014        | 0.073         | 0.014         | 0.098         | 0.018         |
> | **Color3D (Ours)**             | **0.041**        | **0.007**       | **0.062**        | **0.009**        | **0.052**       | **0.008**       |
>
> It is observed that our method consistently achieves optimal consistency, whether for the proposed ME metric or the TC metric. This should further demonstrate the superiority of the proposed approach in simultaneously maintaining consistency across perspectives and over time.
>
> [1] DL3DV-10K: A Large-Scale Scene Dataset for Deep Learning-based 3D Vision. CVPR 2024.
> [2] Learning Blind Video Temporal Consistency. ECCV 2018.

---

> ### Author Response · Authors · 2025-11-21
> **Official Response by Authors -- Part 4**
>
> **Q4: How stable is the personalized colorizer’s output under large camera baselines (e.g., >60° change)? If still good, what are the fundamental factors that may contribute to this effect?**
>
> **A4:** Thanks for your insightful question. Both Mip-NeRF 360 and DL3DV-140 are captured with 180°-360° camera changes, and our method consistently performs well on these challenging scenes. The quantitative and qualitative results demonstrate the effectiveness of our approach under large camera baselines. This robustness can be attributed to four key factors:
>
> - **Data augmentation simulates large viewpoint changes**. We apply large-scale elastic transformations during data augmentation to simulate strong spatial and perspective shifts, enabling the colorizer to robustly handle large-angle camera movements during inference.
>
> - **Frozen encoder brings better generalization.** We utilize a pre-trained encoder capable of extracting robust visual features across diverse scenes and keep its weights frozen while only fine-tuning lightweight adapters. The robustness and stability of the frozen encoder allow our method to generalize more effectively during inference.
>
> - **Generative augmentation simulates unseen content**. We introduce a generative augmentation strategy to synthesize plausible unseen content in the scene while preserving consistent color and style. This enables the colorizer to acquire a more robust ability to handle previously unseen regions and objects. As shown in **Fig. 9 (b)** of the revised paper, the generative augmentation can produce additional scene content with coherent color characteristics, allowing the colorizer to reliably colorize these newly introduced elements.
>
> - **Color interpolation capacity**. Even for completely unseen content, our method can still produce plausible color interpolation. Because the encoder is frozen and trained on extensive datasets to extract robust visual features, our optimization enables the colorizer to maintain consistent features for the same object regardless of spatial position or viewpoint changes, but still preserves the representation capacity of general objects. The decoder trained with key view and augmented images is capable of establishing a relatively broad color mapping and can perform color interpolation for unseen features. As shown in **Fig. 17** of the revised paper, when encountering unseen content, our colorizer is still able to extract meaningful visual features and assign plausible colors by performing color interpolation guided by the learned color mappings.
>
>
>
> **Q5: How sensitive is performance to errors in key-view selection?**
>
> **A5:** Thanks for your valuable question. Our method is robust to key-view selection. As shown in **Fig. 15** of the revised paper, different choices of the key view consistently yield coherent and vibrant 3D colorization results. Even when the first key view contains almost no grass, our approach can still produce reasonable and vibrant colors. Moreover, for distant buildings that do not appear in the key view at all, our method can still infer plausible colors for them. These can be primarily attributed to the generative data augmentation strategy and the colorizer’s ability to interpolate colors for unseen content as described in the above question, which together enable stable colorization even when the selected key view is not ideal.
>
> Additionally, our proposed key-view selection strategy provides a principled mechanism to ensure reliable view selection. By maximizing the feature information entropy, the mechanism robustly selects the view that covers the most diverse scene content across various scenarios, and its effectiveness has been validated in **Tab. 3** of the revised paper.

---

> ### Author Response · Authors · 2025-11-21
> **Official Response by Authors -- Part 5**
>
> **Q6: How could this approach be generalized to feed-forward style 3D generation models?**
>
> **A6:** Thanks for your valuable question. Our method can indeed be extended to a feed-forward style paradigm, and we have already conducted preliminary validation. As illustrated in **Fig. 12** of the revised paper, we first apply our proposed single-view augmentation strategy on a large-scale image dataset to synthesize substantial spatial and viewpoint variations. Using only one colored image as the key view, we train a generalizable personalized colorizer based on a transformer network that learns to colorize all augmented views in a feed-forward manner.
>
> At inference time, we follow the exact pipeline described in the paper: we colorize a single input view and then propagate its chroma to other viewpoints using the generalizable personalized colorizer, without any per-scene optimization. The resulting set of multi-view colorized images can then be fed into a feed-forward 3D reconstruction model—such as AnySplat [3]—to directly obtain a fully colored Gaussian representation. We trained the generalizable personalized colorizer on the COCO dataset and validated the feed-forward 3D colorization pipeline on the DL3DV-140, demonstrating the feasibility of this extension. Tab. R4 reports the quantitative comparison between the generalizable colorizer and our per-scene personalized colorizer for feed-forward 3D reconstruction models, and the corresponding visual results can be found in **Fig. 13** of the revised paper.
>
> Table R4: Quantitative comparisons of automatic colorization on the DL3DV-140 dataset.
> | Method                     | FID↓ | Colorful↑ | ME↓ | TC↓ |
> |-----------------------------|----------------|-------------------|---------------|---------------|
> |  Generalizable Colorizer + AnySplat         |82.36 |24.37|0.142|0.030|
> | Per-Scene Colorizer + AnySplat  |66.78|29.39|0.109|0.025|
>
> Our experiments indicate that the proposed framework already exhibits *promising feed-forward 3D colorization potential*. While a fully feed-forward solution is indeed appealing, our observations show that a globally generalized colorizer still falls short of our per-scene adapter in both chromatic richness and multi-view consistency, and it suffers from significant resolution degradation. Given the large-baseline viewpoints and high-resolution reconstruction settings considered in real-world applications, the per-scene optimization route remains the most practical and reliable choice at present. Nonetheless, this preliminary investigation highlights a viable direction for future research. Due to the limited rebuttal window, we are unable to provide a broader set of experiments at this time. However, we plan to further investigate more effective feed-forward designs for 3D colorization in future work. Thank you again for the insightful question.
>
> [3] AnySplat: Feed-forward 3D Gaussian Splatting from Unconstrained Views. SIGGRAPH Asia 2025.

---

### Author Response · Authors · 2025-11-21
**Brief Summary of the Responses**

Below, we briefly summarize how we have addressed the concerns raised by each reviewer, hoping this will help the new Area Chair quickly capture the core concerns and responses.

**Reviewer uPuV (4)**

- *Generalization ability and large dataset application*. We have demonstrated the generalization ability of our method in **Sec. 4.5**. We have also introduced an effective large-scale training strategy in **Sec. A.4** and provided results on large DL3DV-140 dataset in **Tab. 1**.
- *Add theoretical or diagnostic analysis*. We have provided a comprehensive theoretical analysis in **Sec. A.3** and diagnostic analysis in **Sce. 4.6**.
- *Add larger scale evaluation and temporal consistency metrics*. We have provided more results on the large-scale dataset DL3DV-140 in **Tab. 1** as well as the dynamic dataset HyperNeRF in **Tab. 2**. We have also added a temporal consistency metric to **all Tables**.

**Reviewer Rt9r (6)**

- *Computational cost and practicality for large-scale applications*. We have demonstrated that the computational cost is minimal and further validated the efficiency of our method by showing strong generalization ability in **Sec. 4.5**. We have also introduced an effective large-scale training strategy in **Sec. A.4** and demonstrated the practicality on the large DL3DV-140 dataset in **Tab. 1**.
- *Robustness of single key view scheme*. We have demonstrated the robustness through the resilience to key-view selection (**Fig. 15**), algorithmic guarantees (**Tab. 3**), strategic safeguards (**Fig. 9**), and generalization capability (**Fig. 17**).

**Reviewer Rsix (6)**

- *Add optimization time of the second stage*. We have provided a detailed runtime breakdown in **Tab. 7** and demonstrated our method is more efficient than existing methods.
- *Contribution scope*. We have elaborated on the broader contributions of this work and highlighted its significance and value as **the first framework** to simultaneously achieve controllability, consistency, and color richness in 3D colorization.
- *Add examples on challenging lighting conditions*. We have provided these examples in **Fig. 18** and demonstrated the effectiveness of our method under such conditions.


**Reviewer TQsU (6➡8)**

- *Add comparison with "ChromaDistill" and video colorization baseline*. We have validated the generalization capability in **Sec. 4.5**, proposed an efficient training strategy for large-scale scenarios in **Sec. A.4**, and reported results on the large DL3DV-140 dataset in **Tab. 1**.

- *Add additional view consistency metric*. We have added the metric TC to **all Tables**. Our method still achieves the best performance.

- *Extra training overhead*. We have demonstrated that the additional training overhead is negligible, and further validated the efficiency of our method by showing strong generalization ability in **Sec. 4.5** and introducing an effective training strategy in **Sec. A.4**.

---

> ### Author Response · Authors · 2025-12-01
> **Brief Summary of Paper Revisions**
>
> We sincerely thank all the reviewers for their valuable suggestions. We have revised the paper based on these comments. A brief summary is as follows:
>
> 1. **More experimental results:**
>
>     - Add long-term and short-term temporal consistency (TC) metric across all relevant tables.
>     - Add more comparisons on the 140-scene DL3DV-140 and the dynamic HyperNeRF datasets (Tab. 1 and Tab. 2).
>     - Add video colorization model ColorMNet as a reference baseline (Tab. 1 and Tab. 2).
>     - Add comparisons with ChromaDistill (Tab. 5).
>     - Add visualizations under challenging lighting conditions (Fig. 18).
>
> 2. **More discussion and analysis:**
>
>     - Provide analysis of the generalizability of the personalized colorizer (Line 452).
>     - Provide diagnostic analysis of color consistency in personalized colorizer (Line 463).
>     - Provide rigorous theoretical analysis of color consistency in personalized colorizer (Line 882).
>     - Introduce an all-in-one fine-tuning scheme for large-scale datasets (Line 999).
>     - Introduce a feed-forward 3D colorization paradigm (Line 1019).
>     - Provide analysis of the number of key views (Line 1107).
>     - Provide analysis of the robustness of key view selection (Line 1130).
>     - Provide analysis of generative augmentation for dynamic scenes (Line 1159).
>     - Provide analysis of colorization for semantically similar objects (Line 1184).
>     - Provide analysis of optimization efficiency (Line 1220).

---

### Comment · Area_Chair_Sz58 · 2025-11-26

Dear reviewers,

Please check the author's reply. Feel free to raise any questions or start a discussion, regardless of whether you will change the score.

Your AC.

---

### Author Response · Authors · 2025-12-01
**Summary for the Area Chair**

Dear Area Chairs,

We sincerely thank the reviewers and area chairs for their valuable time and constructive feedback. Below, we provide a brief summary of the review-rebuttal process, which we hope will help inform your decision regarding our submission.

Reviewers **Rt9r**, **Rsix**, and **TQsU** all expressed positive initial assessments of our work with high confidence. In particular, Reviewer **TQsU** further increased the score to **8** after multiple rounds of thorough discussions (22-25 Nov, prior to the data leakage incident). Reviewer **uPuV** also commended our method for its strong user steerability and computational efficiency. Due to the unexpected interruption, Reviewers **Rt9r**, **Rsix**, and **uPuV** were unable to post their final responses. Nevertheless, we have carefully addressed all concerns raised by the reviewers, providing additional experiments, detailed analyses, and new strategies wherever needed. Specifically,

- **Reviewer TQsU (6➡8)** praised the *“technical novelty,” “practical applications,” “controllability,”* and *“thorough experimentation”* of our work. In our rebuttal, we provided additional experiments and thorough analysis, which he/she confirmed that our detailed response addressed all of the concerns and raised the score to **8**. The multiple rounds of discussion and the corresponding dates (22-25 Nov) indicate that this positive consensus was formed based on the merits of our work and our rebuttal, and is entirely unrelated to the information leakage incident.

- **Reviewer Rsix (6)** recognized *“The method is technically sound and well-motivated” and "The experiments are thorough and provide strong empirical validation"*. In our rebuttal, we further supplied a detailed runtime breakdown, added challenging examples, and elaborated on the broader contributions and significance of our framework.

- **Reviewer Rt9r (6)** appreciated *“The paper is well-written and easy to follow”, "... highly practical and valuable for real-world applications", and "... showing superior consistency, color vividness, and alignment with user intent"*. In response, we provided additional demonstrations and thorough analysis, and further substantiated the practicality of our approach with additional experimental evidence.

- **Reviewer uPuV (4)** acknowledged our work for its *“high controllability and flexibility” and “computational practicality”*. Regarding the concerns raised, we provided comprehensive theoretical justifications, additional experimental evidence, and thorough analyses. Furthermore, we introduced a novel training strategy tailored for large-scale applications and validated its effectiveness through new convincing experiments.

Below, we also provide a concise summary of our responses for your convenience. Thank you again for your time and your dedicated service to the community.

Best regards,

Authors of Submission 2907

---

### Meta-Review · Area_Chair_jjFd · 2025-12-31

**Summary:**

This paper was evaluated by four reviewers, receiving an initial mixed response consisting of one borderline rejection and three weak acceptances. The primary concerns raised by the reviewers focused on the generalization ability and scalability constraints inherent to per-scene optimization, the lack of rigorous examination for the inductive bias, and the limited scope of the evaluation datasets, particularly regarding dynamic scenes. Additionally, reviewers questioned the method's sensitivity to key-view selection and requested more direct comparisons with recent state-of-the-art methods like ChromaDistill. The authors provided a comprehensive rebuttal that included new experiments, theoretical analyses, and additional baselines. The rebuttal successfully resolved the majority of the empirical and methodological questions, resulting in a technically sound contribution. Some inherent limitations of the proposed pipeline include scalability and color space formulation. Considering the overall strength of the paper and the substantial rebuttal, I recommend acceptance as a poster.

**Reviewer Concerns:**

The widespread concern regarding the computational cost and scalability of per-scene optimization, raised by most of reviewers, was adequately addressed by the authors through detailed clarifications and an optimization efficiency analysis. Similarly, the critiques regarding the limited evaluation scope, specifically the need for dynamic datasets and comparisons to ChromaDistill raised by Reviewers uPuV and TQsU, were resolved by the inclusion of additional experiments on DL3DV-140 and HyperNeRF datasets, along with the adoption of long-term and short-term view consistency metrics. Technical concerns regarding the sensitivity to key-view selection and the empirical nature of the inductive bias (Reviewers uPuV and Rt9r) were effectively countered with ablation studies and diagnostic analyses. Additionally, the authors demonstrated robustness under challenging lighting conditions. However, the concern raised by Reviewer Rsix regarding the limitation of optimizing in the LAB color space instead of RGB remains only partially addressed, as its generalization across different color space optimizations remains a valid limitation.

**Reviewer Scores:**

Reviewer uPuV initially assigned a score of 4 with a confidence of 4. Given that the rebuttal addressed his/her comprehensive list of concerns, including the addition of dynamic datasets and theoretical justifications, it is estimated that his/her score would likely improve to a 6, moving from rejection to weak acceptance. Reviewer Rt9r, who initially gave a score of 6, would likely maintain this rating or increase the score to 8. Reviewer Rsix assigned a 6 with high confidence (5) and is expected to retain this score; although the lighting and cost concerns were cleared up, the unresolved limitation regarding the LAB vs. RGB color space serves as a valid cap on the score. Reviewer TQsU, who also started with a 6 and high confidence, would likely maintain his/her score or increase it to a 8, as the authors satisfied all requests by integrating the comparison to ChromaDistill and adopting the suggested consistency metrics.

---

### Decision · Program_Chairs · 2026-01-26

Accept (Poster)